# OMNI-RRM: AUTOMATIC PREFERENCE & REASONING CONSTRUCTION ADVANCES MULTIMODAL REWARD MODELING

## ABSTRACT

Multimodal large language models (MLLMs) have shown remarkable capabilities, but their safe deployment is hindered by alignment failures. A critical bottleneck is the lack of effective reward models (RMs), which are typically limited to vision, provide opaque scalar scores, and depend on costly human annotations. To address these challenges, we introduce **Omni-RRM**, the first open-source, reasoning-driven reward model that provides explainable preference judgments across text, image, video, and audio. At the core of our approach is **Omni-Preference**, a novel, large-scale dataset constructed through a fully automated pipeline: we generate preference pairs by contrasting models of varying capabilities and then enrich them with multi-criteria, chain-of-thought rationales from powerful teacher models, completely eliminating the need for human labeling. Omni-RRM is trained in a two-stage process: supervised fine-tuning to instill the ability to generate structured rationales, followed by reinforcement learning to sharpen its judgment on difficult, low-contrast examples. Comprehensive evaluations show that Omni-RRM achieves state-of-the-art accuracy on video (80.2% on ShareGPT-V) and audio (66.8% on Audio-HH) benchmarks, while significantly outperforming existing open-source RMs on image tasks, with an overall improvement of 17.7% over its base model. Furthermore, Omni-RRM demonstrates strong generalization, boosting downstream task performance via Best-of-N selection and even improving accuracy on text-only preference benchmarks. Our data, code and models are available at `https://anonymous.4open.science/r/Omni-RRM-CC08`

## 1 INTRODUCTION

As Multi-modal Large Language Models (MLLMs) like GPT-4o, Gemini 1.5, and Qwen2.5-Omni achieve unprecedented capabilities in processing real-time, multi-stream data (OpenAI, 2024; Gemini Team, 2024; Xu et al., 2025c), a critical chasm has emerged. While their raw power continues to soar, ensuring their outputs align with human values has become the most significant challenge for their safe and reliable real-world deployment. Foundational models demonstrate remarkable perception and reasoning (Liu et al., 2023; Chen et al., 2023; Bai et al., 2023), yet a persistent gap remains between their capabilities and their reliability. This alignment problem is critically bottlenecked by the current generation of reward models (RMs), which are fundamentally ill-equipped for the complexities of the multimodal world and represent the Achilles' heel of trustworthy AI.

Existing RMs suffer from a trifecta of crippling limitations that collectively stifle progress. First, they have **incomplete modality coverage**. The overwhelming research focus on vision leaves crucial modalities like audio almost entirely unevaluated (Wang et al., 2023), creating a significant blind spot in our ability to align models that perceive the world as humans do. Second, they are **critically opaque**. The standard practice of providing a single, uninformative scalar score offers no insight into *why* one response is preferred, hindering user trust, preventing targeted debugging, and making iterative improvement a matter of guesswork (Kirstain et al., 2023; Xiong et al., 2024a). Third, their development is **prohibitively expensive and slow**. The reliance on manual human annotation for methods like Reinforcement Learning from Human Feedback (RLHF) chains progress to a costly, non-scalable, and often inconsistent data labeling process, limiting adaptation to new

domains(Kaufmann et al., 2023). This triad of flaws makes robust, comprehensive alignment an unsolved and pressing challenge.

From a game-theoretic perspective, the alignment challenge can be viewed as a principal–agent dilemma. Researchers act as principals, aiming to instill nuanced human preferences into a reward model agent. Yet the agent cannot be directly controlled; instead, it must be guided through incentive mechanisms. Existing reward models correspond to weak mechanisms: they provide only low-information scalar signals, lack transparency into the reasoning behind judgments, and incur unsustainable costs due to human annotation. In this framing, the core task is mechanism design—constructing a scalable and explainable reward process that induces the agent to reach an equilibrium behavior faithfully aligned with the principal's intent.

To dismantle these fundamental barriers, we introduce **Omni-RRM**, an open-source, reasoning-driven reward model designed from the ground up for comprehensive, scalable, and transparent multimodal alignment. Our approach is built on two core innovations designed to directly address the limitations of prior work: a self-sustaining, fully automated data engine and a progressive training strategy that unlocks the model's latent reasoning abilities.

The foundation of our work is the **Omni-Preference Dataset**, a large-scale corpus of preference pairs spanning images, video, and audio that decouples alignment from the human annotation bottleneck. We construct it via a novel, zero-human-annotation pipeline: preference labels are first automatically derived by contrasting outputs from models with a known 'capability-gap'. These pairs are then enriched with structured, multi-criteria rationales from powerful teacher models, creating a rich supervisory signal. Built upon this data, Omni-RRM is trained in a synergistic two-stage regimen. Supervised Fine-Tuning (SFT) first teaches the model to imitate the structure of a high-quality critique, generating an explicit chain-of-thought rationale before making a judgment. This is followed by Group Relative Policy Optimization (GRPO), a reinforcement learning phase that sharpens its decision-making on nuanced and challenging cases where simple imitation fails. This synergy produces a reward model that is not only highly accurate but also inherently interpretable.

Our extensive evaluations demonstrate the superiority of the Omni-RRM paradigm. It sets a new state-of-the-art on video (80.2% on ShareGPT-V) and audio (66.8% on Audio-HH) benchmarks, domains where prior open-source RMs could not even compete. It also surpasses all open-source baselines on established image tasks, achieving a remarkable overall performance gain of 17.7% over its unaligned backbone. Crucially, Omni-RRM proves its practical utility as an alignment tool, consistently improving inference-time performance across various models via Best-of-N selection. Furthermore, it exhibits strong generalization, enhancing accuracy even on text-only preference tasks—a powerful confirmation that the principles of reasoned judgment it learns are modality-agnostic. Omni-RRM represents a crucial step towards building truly reliable and trustworthy multimodal AI systems, closing the critical gap between raw capability and value-aligned behavior.

In summary, our contributions are three-fold:

- A novel, fully automated preference dataset named **Omni-Preference** that spans image, video, and audio, enriched with multi-criteria Chain-of-Thought rationales and rigorously filtered for high confidence.
- A strong unified reward model **Omni-RRM** supporting four modalities and generating explicit, reasoned judgments, trained via a two-stage SFT + GRPO regimen with carefully designed reward functions.
- Extensive evaluations across four modalities demonstrating the state-of-the-art preference accuracy of Omni-RRM, significant gains over strong baselines, and effective downstream alignment through Best-of-N inference as well as text-only generalization.

## 2 RELATED WORK

### 2.1 MULTI-MODAL LARGE LANGUAGE MODELS

Recent years have seen rapid advancements in open-source MLLMs, evolving from foundational vision-language tuning in LLaVA (Liu et al., 2023; 2024a; Li et al., 2024) to billion-parameter backbones in InternVL (Chen et al., 2023; Zhu et al., 2025), and native audio-visual processing in series like Qwen-VL (Bai et al., 2023; 2025; Xu et al., 2025a). Cutting-edge models now tackle

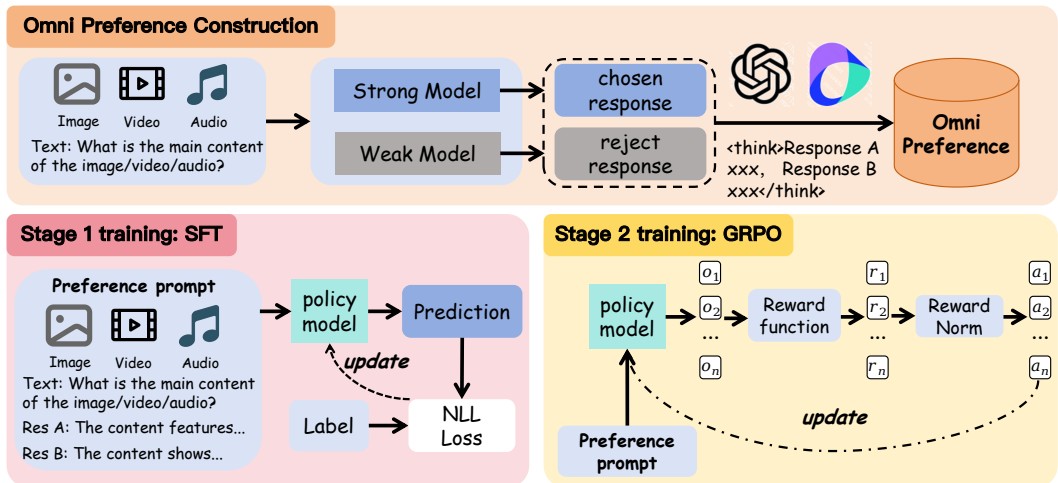

Figure 1: The overall pipeline for creating Omni-RRM. The process consists of two main phases. **Top: Omni Preference Construction.** Preference pairs are first automatically generated by contrasting outputs from a strong and a weak model. Powerful teacher models then annotate these pairs with detailed rationales to create the final dataset. **Bottom: Two-Stage Training.** In Stage 1 (SFT), the policy model is trained on preference prompts using a standard NLL loss. In Stage 2 (GRPO), the model is further refined by generating multiple responses, scoring them with a rule-based reward function, and updating the policy based on the normalized reward.

complex reasoning over long-form content (Feng et al., 2025; Yang et al., 2025). However, this explosion in capabilities has created a critical gap, as the alignment techniques required to ensure their safety and reliability have not evolved at the same pace.

## 2.2 PREFERENCE ALIGNMENT

Preference alignment is typically achieved via Reinforcement Learning from Human Feedback (RLHF) (Kaufmann et al., 2023), with recent work exploring automated AI feedback (RLAIF) (Lee et al., 2023) and more direct policy optimization methods like DPO and GRPO (Rafailov et al., 2023; Liu et al., 2024c) to reduce complexity. However, extending these successes to multimodal domains is non-trivial (Wang et al., 2023; Liu et al., 2024b). The quality of the underlying reward model (RM) becomes the central bottleneck (Metz, 2024), as it must provide a reliable learning signal across diverse and complex data streams. To overcome these limitations, we propose **Omni-Preference**, a fully automated data synthesis pipeline, and leverage *rubric-grounded* GRPO to train an **Omni-Reward Model**. This approach moves beyond scalar reward signals by optimizing a *composite* objective (format, preference, rationale), enabling fine-grained, controllable preference modeling and interpretable supervision across text, image, video, and audio.

## 2.3 MULTIMODAL REWARD MODELS

However, the development of multimodal RMs has significantly lagged behind the generative models they are meant to align. The prevailing paradigm faces several critical gaps. **First, existing models exhibit a strong bias towards visual modalities (image and video).** This visual-centric approach is evident across the literature, like ImageReward (Xu et al., 2023), Pick-a-Pic (Kirstain et al., 2023) and BaseReward (Zhang et al., 2025a), which concentrates its analysis on visual tasks. This leaves a significant void in the evaluation of other crucial data streams, particularly **audio**. **Second, they often lack interpretability.** Models like LLaVA-Critic (Xiong et al., 2024a) may generate textual feedback, but it is often unstructured, with the dominant approach relying on producing a simple scalar score. Despite the high performance achieved on specific benchmarks, the fundamental reliance on pre-existing data sources and the lack of structured, interpretable reasoning remain key limitations of this paradigm. In contrast, our work with Omni-RRM is designed to holistically address these challenges by introducing a fully automated data synthesis pipeline, an

inherently interpretable reasoning process, and a truly omni-modal scope that includes dedicated audio support.

Recently, to address the interpretability limitations of scalar scores, an emerging paradigm has shifted towards augmenting RMs with explicit Chain-of-Thought (CoT) reasoning. For instance, UnifiedReward-Think (Wang et al., 2025b) introduces a multimodal CoT reward model for vision tasks, trained via distillation and GRPO-style reinforcement fine-tuning on existing datasets. VR-Thinker (Wang et al., 2025a) extends this reasoning capability to long-form video by leveraging visual memory mechanisms. In the text domain, RM-R1 (Chen et al., 2025) frames reward modeling as a reasoning problem, training generative RMs through distillation combined with verifiable RL. Omni-RRM advances this direction but distinguishes itself through three structural innovations: (1) **Autonomous Data Construction**: Rather than relying on pre-existing preference-labeled corpora, we construct Omni-Preference via a fully automated, label-free pipeline driven by strong–weak model contrast and multi-teacher rubric annotation (§4.1); (2) **Fine-Grained Reward**: Instead of coarse outcome-based rewards, we optimize a composite, rubric-grounded objective that enforces structured comparative reasoning across five evaluation dimensions (§4.2); and (3) **Omni-Modal Support**: While prior CoT RMs are restricted to text or vision/video domains, Omni-RRM unifies text, image, video, and audio within a single reward model.

## 3 PROBLEM FORMULATION

Traditional reward models (RMs) formulate the alignment problem as a regression task. Given a context $x$ and a response $y$, the model $r_\theta(x, y)$ is trained to output a scalar score $s \in \mathbb{R}$, which represents the quality of the response. While computationally efficient, this formulation is fundamentally opaque, offering no insight into the model's decision-making process.

In contrast, we reformulate reward modeling as a **structured generation task**, shifting the paradigm from producing opaque scores to generating interpretable, reasoned judgments. Instead of a scalar, our model, which we denote as a policy $\pi_{\text{RRM}}$, is trained to generate a complete textual output $Y$ that explains its preference.

Given a multi-modal context $x$ and a pair of candidate responses $(y_A, y_B)$, the generation process is defined as:

$$Y = \pi_{\text{RRM}}(\phi(x, y_A, y_B)) \tag{1}$$

where $\phi$ is a function that formats the inputs into a prompt. The structured output $Y$ is a concatenation of two sequential components:

1. **A Chain-of-Thought (CoT) Rationale** ($T$)**:** The first part of the output is an analytical critique that evaluates the merits and demerits of each candidate response. This rationale provides a transparent, interpretable foundation for the final decision.

2. **A Final Judgment** ($l^*$)**:** The output culminates in a discrete token, $l^* \in \{A, B\}$, which explicitly identifies the preferred candidate. This judgment serves as the direct preference signal for downstream alignment.

This formulation transforms the reward model from an opaque scalar predictor into an interpretable reasoning agent, providing a richer and more transparent signal for alignment.

## 4 METHOD

To realize our formulation of reasoned judgments, we introduce a holistic methodology centered on two core contributions, as depicted in Figure 1. First, to provide the necessary supervision for this structured generation task, we construct the **Omni-Preference** dataset through a novel, fully automated data synthesis pipeline. Second, we develop a progressive two-stage training strategy for **Omni-RRM** that combines Supervised Fine-Tuning (SFT) to instill foundational skills with Group Relative Policy Optimization (GRPO) to sharpen its fine-grained judgment.

Table 1: Detailed statistics of the synthesized **Omni-Preference** dataset. We list the **Strong** ($M_S$) and **Weak** ($M_W$) models used in our capability-gap generation, along with difficulty distribution and final sample counts. **Difficulty definition:** a pair is labeled *Hard* if $|\text{score}_A - \text{score}_B| < 2$, and *Easy* otherwise ($\geq 2$).

| Modality | Data Source | Strong Model ($M_S$) | Weak Model ($M_W$) | Difficulty (Hard / Easy) | Samples |
|---|---|---|---|---|---|
| Image | RLAIF-V | Qwen2.5-VL-7B | Qwen2.5-VL-3B | 5.3k / 11.7k | 17.0k |
| | | Qwen2.5-VL-7B | LLaVA-1.5-7B | | |
| Video | ActivityNet, Charades, Ego4D, NextQA, YouCook2 | Qwen2.5-VL-7B | Qwen2.5-VL-3B | 3.3k / 8.9k | 12.2k |
| Audio | Clotho-AQA | R1-AQA-7B | Qwen2-Audio-7B | 3.0k / 8.8k | 11.8k |
| | | Qwen2.5-Omni-7B | Qwen2-Audio-7B | | |
| | | Qwen2.5-Omni-7B | Qwen2.5-Omni-3B | | |
| **Total** | | | | **11.6k / 29.4k** | **41.0k** |

### 4.1 OMNI-PREFERENCE: A REASONING-AUGMENTED DATASET FOR OMNI-MODAL REWARD MODELING

A reasoning-driven reward model requires a large-scale dataset of preference pairs enriched with explanatory rationales. To create this without costly human annotation, we introduce **Omni-Preference**, a high-quality corpus built via a novel, fully automated two-stage procedure. A full breakdown is provided in Table 1.

**Stage 1: Automated Preference Pair Generation.** First, we generate raw preference pairs using a 'capability-gap' approach. We prompt a strong model ($M_S$) and a weak model ($M_W$) with the same input query ($I, Q$) from public benchmarks such as RLAIF-V (Yu et al., 2025), ActivityNet (Caba Heilbron et al., 2015), and Clotho-AQA (Lipping et al., 2022). Assuming the stronger model produces a better response, we automatically label the pair without human intervention:

$$(r_{\text{acc}}, r_{\text{rej}}) = (M_S(I, Q), M_W(I, Q)) \tag{2}$$

**Stage 2: Rationale Annotation and Filtering.** In this stage, we enrich the preference pairs with structured rationales using two teacher models (GPT-4o-mini and doubao-1.5-pro), providing numerical scores and multi-criteria rationales (fluency, relevance, accuracy, etc.). To ensure high-quality annotations, we apply a robust filtering process that reconciles annotations from multiple teachers and discards inconsistent or low-quality samples. To mitigate bias, we only retain annotations with high teacher agreement. Disagreements are reconciled through score averaging and rationale merging, with a small fraction of conflicting samples manually reviewed to prevent error propagation. Rationale quality is assessed primarily through automated checks on dimensions like fluency, relevance, and accuracy, supplemented by partial manual review. Substandard rationales are discarded, yielding a high-confidence dataset of 41K preference examples with minimal systematic bias. While our dataset is moderate in scale, prior research indicates that reward modeling benefits significantly more from data quality than sheer quantity (Yu et al., 2024a; Liang et al., 2024; Ding et al., 2025; Yu et al., 2024b). We explicitly prioritize high-quality data curation, with complete specifications of our reconciliation policy and filtering pipeline detailed in Appendix A.4. To empirically validate this design choice, we conduct an ablation study comparing Omni-Preference against LLaVA-Critic under an identical SFT+RL setup. The results, shown in Appendix A.2, confirm the effectiveness of our automated pipeline despite the smaller data scale. We leave the exploration of larger-scale training for future work.

### 4.2 TRAINING OMNI-RRM: A PROGRESSIVE SFT-RL APPROACH

Omni-RRM is trained with a progressive two-stage strategy designed to first build a strong reasoning foundation and then refine its judgment. We train Omni-RRM as a *pairwise* comparative reward

model; it is not calibrated to output absolute quality scores for isolated responses, and our evaluations therefore focus on relative preference accuracy. The initial Supervised Fine-Tuning (SFT) stage instills the foundational skills of generating structured, rationale-based evaluations. The subsequent Reinforcement Learning (RL) stage then sharpens its discrimination on nuanced and difficult cases where simple imitation learning falls short. We use a bounded composite objective that combines preference correctness, rubric-grounded rationale quality, and a small structural format term (Appendix A.1). The format term is purely syntactic; once JSON validity and required fields are satisfied, it becomes constant and does not yield further gains, preventing the critic from optimizing format at the expense of reasoning. Structured-output constraints of this kind are widely used to stabilize critics without biasing semantics (Wang et al., 2024; Lu et al., 2025).

**Stage 1: Supervised Fine-Tuning (SFT).** The primary objective of the SFT stage is to teach the model the fundamental structure of our task: generating a coherent rationale followed by a definitive judgment. This phase acts as a form of imitation learning, where the model learns the syntax and style of a high-quality critique from the **Omni-Preference** dataset. To achieve this efficiently, we employ lightweight LoRA fine-tuning (Hu et al., 2022), adapting the pre-trained backbone without incurring the cost of a full fine-tune. The model is trained to autoregressively generate the ground-truth rationale and final verdict for each preference pair. This process is optimized by minimizing the standard negative log-likelihood (NLL) loss over the sequence of target tokens:

$$\mathcal{L}_{\text{SFT}}(\theta) = -\sum_{i=1}^{T} \log P_\theta(y_i \mid y_{<i}, x) \tag{3}$$

This stage is critical as it provides a robust initialization for the subsequent optimization phase. By the end of SFT, the model has already learned the required output format and the vocabulary for comparative evaluation, ensuring that the RL stage begins from a high-quality policy rather than exploring from a random or unaligned state.

**Stage 2: Reinforcement Learning with GRPO.** We use large-scale, cross-modal SFT primarily to teach the rubric schema and stabilize generation before GRPO. Without this step, reinforcement learning tends to spend updates repairing format/structure rather than calibrating *low-margin* preferences; a structured cold start lets GRPO focus on fine-grained comparative reasoning (Paul, 2025; Lu et al., 2025). While SFT is effective for imitation, it cannot teach the model to make fine-grained distinctions beyond the explicit examples in the training set. To sharpen the model's judgment and improve its performance on more ambiguous, low-contrast examples, we employ a reinforcement learning stage using Group Relative Policy Optimization (GRPO) (Liu et al., 2024c). We design a composite reward function $r$ that provides a holistic evaluation signal for each generated output:

$$r = w_{\text{fmt}} \cdot R_{\text{fmt}} + w_{\text{pref}} \cdot R_{\text{pref}} + w_{\text{rub}} \cdot R_{\text{rub}} \tag{4}$$

Each component of the reward function is designed to assess a distinct aspect of output quality:

- **Format Correctness ($R_{\text{fmt}}$):** This component provides a strict, binary reward that ensures the model's output adheres to the required JSON structure. It acts as a guardrail to prevent the policy from drifting into generating malformed outputs during exploration.
- **Preference Accuracy ($R_{\text{pref}}$):** This is the core learning signal, rewarding the model for correctly identifying the winning candidate and for producing scores that align with the ground-truth labels. This directly incentivizes the model to improve its judgment.
- **Rationale Quality ($R_{\text{rub}}$):** To prevent the model from "reward hacking" by producing a correct verdict with a trivial rationale, this component programmatically evaluates the quality of the textual explanation. It rewards the coverage of pre-defined rubric criteria (e.g., fluency, relevance) and the use of comparative language, encouraging the generation of substantive, insightful reasoning.

The detailed formulation for each component and their respective weights are provided in Appendix A.1. The policy is then optimized by minimizing the standard GRPO loss. This loss function normalizes the rewards for a group of $k$ generated samples, computing a normalized advantage $A_i$ for each response that indicates its quality relative to the group average.

$$\mathcal{L}_{\text{GRPO}}(\theta) = -\frac{1}{k}\sum_{i=1}^{k} A_i \cdot \log \pi_\theta(y_i \mid x_i), \quad \text{where } A_i \text{ is the normalized advantage.} \tag{5}$$

Table 2: Main results on multi-modal preference benchmarks. We report accuracy (%) across benchmarks, with a focus on preference-based evaluation. The **best** in each column is bold and the second best is underlined. "—" means not evaluated for that modality. **Overall** is the mean accuracy across supported modalities; for models lacking a modality (e.g., audio), the mean is taken over available ones. Values in parentheses are relative improvements (%) over the corresponding Qwen2.5-Omni baselines.

| Model | VL-Reward | MM-RewardBench | ShareGPT-Video | Audio-HH | Overall |
|---|---|---|---|---|---|
| *Proprietary models* | | | | | |
| GPT-4o-mini | 59.8 | 61.9 | 53.9 | 58.2 | 58.4 |
| Doubao-1.5-Vision-Pro | 77.3 | 68.0 | 77.0 | — | — |
| Gemini-2.0-Flash | 73.4 | 62.8 | 74.6 | 60.1 | 67.7 |
| Gemini-2.5-Pro | **79.6** | 63.3 | 78.8 | 66.5 | **72.0** |
| *Open-source models* | | | | | |
| Qwen2.5-Omni-3B | 53.7 | 53.9 | 58.1 | 58.7 | 56.1 |
| Qwen2.5-Omni-7B | 57.8 | 57.5 | 66.3 | 62.4 | 61.0 |
| Qwen2.5-VL-3B | 53.2 | 53.3 | 61.2 | — | — |
| Qwen2.5-VL-7B | 58.2 | 56.0 | 70.5 | — | — |
| Qwen2.5-VL-72B | 62.3 | 63.5 | 72.9 | — | — |
| *Open-source reward models* | | | | | |
| LLaVA-Critic-7B | 54.1 | 56.0 | — | — | — |
| Skywork-VL-Reward-7B | 60.4 | 67.4 | 59.9 | — | — |
| UnifiedReward-think-7B | 66.6 | 71.4 | 77.8 | — | — |
| R1-Reward-7B | 65.8 | 72.3 | 58.7 | — | — |
| **Ours** | | | | | |
| Omni-RRM-3B (*sft*) | 56.8 | 58.1 | 64.9 | 60.3 | 60.0 |
| Omni-RRM-7B (*sft*) | 60.4 | 61.0 | 70.5 | 62.8 | 63.7 |
| Omni-RM-3B (sft+rl) | 56.9 | 54.7 | 60.0 | 63.8 | 58.9 |
| Omni-RRM-3B (*sft+rl*) | 58.5 (+8.9%) | 68.9 (+27.8%) | 67.4 (+16.0%) | 65.1 (+10.9%) | 65.0 (+15.9%) |
| Omni-RM-7B (sft+rl) | 64.0 | 60.6 | 68.7 | 64.3 | 64.4 |
| Omni-RRM-7B (*sft+rl*) | 67.1 (+16.1%) | **72.9** (+26.8%) | **80.2** (+21.0%) | **66.8** (+7.1%) | 71.8 (+17.7%) |

This progressive two-stage training regimen ensures that Omni-RRM first learns the structure of reasoned evaluation and then refines its ability to make discerning judgments, resulting in a more robust and accurate reward model.

# 5 EXPERIMENTS

## 5.1 BENCHMARKS AND METRICS

We evaluate Omni-RRM on a diverse suite of preference benchmarks spanning three modalities. For all benchmarks, our primary metric is **Preference Accuracy**, which measures the model's alignment with ground-truth human judgments.

**Image:** For image evaluation, we use two standard benchmarks. **VL-RewardBench** (Li et al., 2025) assesses reasoning depth, and we report performance on its official reasoning accuracy metric. **Multi-modal RewardBench** (Yasunaga et al., 2025) is a comprehensive benchmark with rigorous human annotations; we report results on its primary Preference metric.

**Video:** To evaluate video understanding, we adapt the **ShareGPT-Video DPO** dataset (Zhang et al., 2024). We construct a preference benchmark by selecting pairs of human-rated responses with a significant score difference, ensuring clear ground-truth labels.

**Audio:** For audio, we create a synthetic benchmark named **Audio-HH-RLHF**. We convert the text prompts from the well-established *HH-RLHF* corpus (Bai et al., 2022) into speech using a TTS engine, while preserving the original human-vetted preference labels as ground truth. We acknowledge the limitations of this synthetic approach, as it lacks real-world recording noise, speaker variation, and prosody artifacts. And currently, we have not identified a suitable benchmark for real audio preferences.Experiments on authentic audio data remain a key direction for future work.

## 5.2 BASELINES

We benchmark Omni-RRM against a comprehensive set of baselines. First, we establish a performance ceiling using leading proprietary MLLMs such as GPT-4o-mini (OpenAI, 2024) and Gemini-2.0-Flash (Google DeepMind, 2025). Second, we compare against general-purpose open-source MLLMs, particularly the Qwen-2.5 series (Xu et al., 2025b); as our backbone, the original Qwen-2.5-Omni serves as a direct unaligned baseline to measure our training gains. Third, we evaluate against direct competitors in the form of specialized reward models, including LLaVA-Critic-7B (Xiong et al., 2024b), R1-Reward-7B (Zhang et al., 2025b), and UnifiedReward-think-7B (the Qwen-based variant) (Wang et al., 2025b). Finally, for a rigorous internal ablation, we compare our full model, Omni-RRM (with rationale supervision), against Omni-RM (without rationale supervision), a no-reasoning reward model trained under the identical SFT+RL pipeline.

## 5.3 EXPERIMENTAL RESULTS

Our main experimental results, presented in Table 2, lead to three key conclusions.We further analyze low-contrast preference scenarios by stratifying test pairs into hard/easy categories based on teacher score margins. Omni-RRM brings consistently larger gains on hard pairs; see Appendix B.1.

**Omni-RRM is the Premier Open-Source Reward Model, Rivaling Top Proprietary Systems.** Our Omni-RRM-7B establishes itself as the leading open-source reward model by setting a new state-of-the-art in both video and audio domains, surpassing even the top proprietary model, Gemini-2.5-Pro. It achieves an accuracy of 80.2% on ShareGPT-V (video) and 66.8% on Audio-HH (audio). This strong cross-modal performance culminates in an Overall score of 71.8%, placing it just 0.2 points behind Gemini-2.5-Pro and solidifying its position as the best-performing open-source model. Concurrently, it remains highly competitive on image benchmarks, securing the highest score among all models on MM-RewardBench (72.9%).We further compare an audio-only RM with an omni RM under identical pipelines; the omni setup yields +2.1 pp on Audio-HH-RLHF (Appendix B.3, Table 7).

**Progressive Training Unlocks Base Model Potential.** Our two-stage progressive training strategy, combining Supervised Fine-Tuning (SFT) with Group Relative Policy Optimization (GRPO), provides a substantial performance uplift over the unaligned base models. For instance, after this training, our Omni-RRM-7B model demonstrates remarkable gains over its Qwen-2.5-Omni-7B backbone: accuracy soars by +15.4 absolute points on MM-RewardBench (a +26.8% relative improvement) and +13.9 points on ShareGPT-V (a +21.0% relative improvement). This highlights the synergy of our approach: the initial SFT stage instills foundational knowledge, while the subsequent GRPO stage sharply hones the fine-grained reasoning required for complex preference modeling.

**Specialized Training Outweighs Raw Model Scale.** A key finding is that reward modeling is a distinct capability where targeted training can outperform even the most powerful general-purpose MLLMs. This is evident on the MM-RewardBench, where a top-tier generalist like Gemini-2.5-Pro (63.3%) is significantly outperformed by specialized open-source reward models, including our Omni-RRM-7B (72.9%), R1-Reward-7B (72.3%), and UnifiedReward-think-qwen-7B (71.4%). This indicates that high-quality preference modeling is a specialized skill that benefits more from targeted, in-domain training data and alignment techniques than from general pre-training alone.

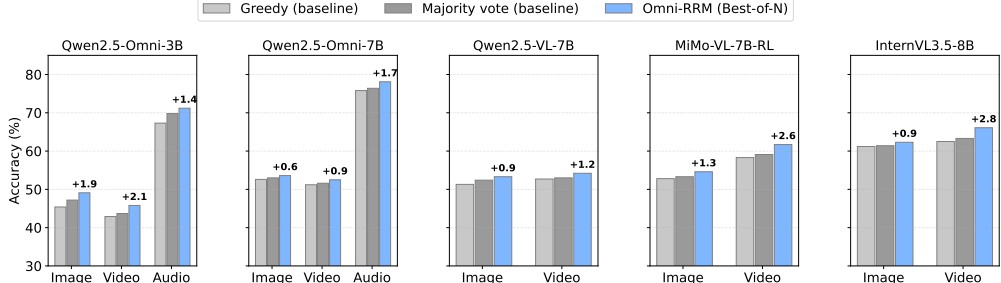

Figure 2: Average accuracy (%) across available modalities (Image, Video, Audio) for five models. We compare **Greedy** (single decoding), **Self-consistency** (majority vote over $N=5$ generations), and our **Omni-RRM re-ranking**, which generates $N=5$ candidates and selects the final output via *pairwise* preference comparisons among them. Omni-RRM consistently improves over both baselines across all models. (Values above the Omni-RRM bars denote improvements over Self-consistency.)

**Explicit Reasoning is a Key Driver of Performance.** A direct comparison between our full model, **Omni-RRM**, and its no-reasoning counterpart, **Omni-RM**, reveals that rationale supervision is not merely for interpretability but is a critical driver of performance. With identical backbones and training pipelines, Omni-RRM-7B substantially outperforms Omni-RM-7B in Overall accuracy (71.8% vs. 64.4%), with the margin becoming even more pronounced on reasoning-intensive benchmarks like MM-RewardBench (72.9% vs. 60.6%). This demonstrates that forcing the model to generate a structured, step-by-step rationale introduces a powerful inductive bias, compelling it to develop a more robust and generalizable internal model of human preferences.

## 5.4 INFERENCE ALIGNMENT VIA BEST-OF-N

To validate the practical utility of Omni-RRM as a plug-and-play alignment tool, we employ a Best-of-N (BoN) re-ranking methodology at inference time. The process is straightforward: for any given query, a base model generates $N = 5$ candidate responses. Our **Omni-RRM-7B** then acts as an expert judge, performing pairwise comparisons among all candidates to select the optimal one as the final answer. This method requires no additional training of the base models. We apply it across five different backbones on three diverse multimodal benchmarks: MMMU (image) (Yue et al., 2024), Video-MME (video) (Fu et al., 2025), and AVQA (audio-visual) (Yang et al., 2022). Performance is benchmarked against two strong baselines: standard greedy decoding and self-consistency (a majority vote over the same $N = 5$ candidates).

The results, summarized in Figure 2, demonstrate that Omni-RRM consistently and effectively improves the performance of all tested models. The gains are evident across different model sizes and architectures. For example, re-ranking with Omni-RRM boosts the accuracy of the Qwen2.5-Omni-7B model by +2.7 points on MMMU and +2.3 on AVQA. Critically, our BoN approach consistently surpasses not only greedy decoding but also the strong self-consistency baseline across the board (e.g., on Video-MME: 52.5% vs. 51.6% for the 7B model). This confirms that Omni-RRM provides a robust and generalizable reward signal, serving as an effective mechanism for inference-time alignment across various models and modalities. We report 95% confidence intervals (paired bootstrap; $N=5$ with Qwen2.5-Omni-7B as the generator) and a latency/gain analysis in Appendix B.5–B.6.

## 5.5 GENERALIZATION TO TEXT-ONLY REWARD MODELING

A critical question is whether our multimodal alignment strategy compromises the model's foundational text-based reasoning capabilities. To investigate this, we evaluate Omni-RRM on the **HH-RLHF** Bai et al. (2022) benchmark, a purely textual preference task, with results presented in Figure 3.

The evaluation reveals that our training pipeline not only preserves but significantly enhances text-only performance. Our **Omni-RRM-7B** model improves its accuracy over its backbone from 67.6%

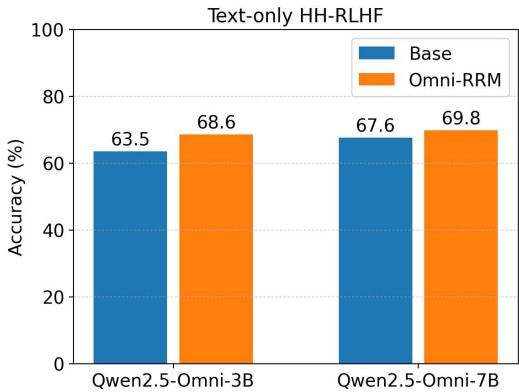

Figure 3: Accuracy (%) on the text-only HH-RLHF benchmark. Omni-RRM consistently improves over its baselines, confirming that multi-modal alignment generalizes effectively to text-only preference modeling without sacrificing core text skills.

to 69.8% (+2.2), and the effect is even more pronounced at the 3B scale, with an improvement from 63.5% to 68.6% (+5.1). These consistent gains across model sizes demonstrate a robust positive transfer effect: the principles of reasoned judgment learned during multimodal alignment successfully generalize, strengthening the model's core text-based preference evaluation abilities. Beyond backbone gains, we also compare against recent text-preference baselines on HH-RLHF (including R1-Reward and UnifiedReward-think); see Appendix B.4, Table 8.

## 6 CONCLUSION

In this paper, we introduced a scalable solution to advance multimodal alignment. Our core contributions are twofold: **Omni-Preference**, a novel dataset created via a fully automated, zero-human-annotation pipeline, and **Omni-RRM**, a reasoning-driven reward model trained upon it. Trained with a progressive SFT+GRPO strategy, Omni-RRM learns to generate interpretable, reasoned judgments across images, video, and audio. Extensive experiments validate our approach. Omni-RRM achieves state-of-the-art accuracy on video (80.2% on ShareGPT-V) and audio (66.8% on Audio-HH) benchmarks, with its overall performance rivaling top proprietary systems. We also confirmed its practical utility, showing it serves as an effective reward signal for improving downstream alignment via Best-of-N selection and even generalizes to text-only tasks. By providing a scalable data framework and a powerful, transparent evaluation model, our work offers a robust pathway for aligning the next generation of powerful and trustworthy MLLMs.

## 7 ETHICS STATEMENT

We have carefully considered the ethical implications of this work, focusing on data sourcing, potential biases, and responsible model use.

**Data Sourcing and Automation.** Our Omni-Preference dataset is constructed via a fully automated pipeline using public academic benchmarks and model-generated responses. This 'zero-human-annotation' approach is a deliberate design choice to mitigate the ethical risks associated with large-scale data labeling, such as unfair compensation and exposure to harmful content. The data is intended for non-commercial research purposes only.

**Potential Biases.** We acknowledge that our automated pipeline inherits and propagates biases from its sources, including the datasets and the foundation models used for generation and annotation. Consequently, Omni-RRM's preference judgments are not a source of objective truth but a reflection of its training data. We urge users to be mindful of these inherent biases and encourage further research into their mitigation.

**Intended Use and Misuse.** Omni-RRM is intended as an open-source research tool to improve the alignment, safety, and interpretability of MLLMs. While we recognize the potential for misuse (e.g., fine-tuning on malicious data to align models toward harmful behaviors), we believe that open-sourcing our entire framework fosters the transparency and community scrutiny necessary for developing more robust and equitable alignment techniques.

## 8 REPRODUCIBILITY STATEMENT

We are fully committed to ensuring the reproducibility of our research. To facilitate the verification of our findings and encourage future work, we publicly release the **Omni-Preference dataset**, our **training and evaluation code**, and the final weights for the **Omni-RRM** model at `https://anonymous.4open.science/r/Omni-RRM-CC08`.

The main body of our paper and its appendices provide a comprehensive description of our methodology. Specifically:

- **Dataset Construction:** The fully automated pipeline for creating the Omni-Preference dataset is detailed in Section 4.1, including the specific data sources, the "capability-gap" model pairs, and the teacher models used for rationale generation.
- **Training Protocol:** Appendix A.3 outlines the key hyperparameters and settings for both the Supervised Fine-Tuning (SFT) and Group Relative Policy Optimization (GRPO) stages, including LoRA configurations, learning rates, batch sizes, and hardware specifications. The specific reward function for the GRPO stage is detailed in Appendix A.1.
- **Evaluation and Prompts:** Our evaluation methodology, including frame sampling rates and decoding parameters, is described in Appendix A.3. To ensure exact replication, all prompts used for data synthesis, training, and inference are provided in Tables 8 through 11 in the Appendix.

We are confident that the public release of our dataset, code, and models, combined with the detailed documentation provided, will allow the research community to easily reproduce our results and build upon our contributions.

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

# A  APPENDIX

## A.1  REWARD FUNCTION DETAILS

This appendix provides detailed descriptions of the three reward components used in our reinforcement optimization framework. Beyond the three components, we also specify the *composite objective* used by GRPO so the overall signal is bounded and stable:

$$R_{\text{total}} = \lambda_{\text{pref}} R_{\text{pref}} + \lambda_{\text{rub}} R_{\text{rub}} + \lambda_{\text{fmt}} R_{\text{fmt}}, \quad R_{\text{total}} \in [-1, 1],$$

with default weights $(\lambda_{\text{pref}}, \lambda_{\text{rub}}, \lambda_{\text{fmt}}) = (0.5, 0.3, 0.2)$. Each sub-reward is clipped to $[-1, 1]$ before aggregation to improve training stability.

### A.1.1 FORMAT REWARD ($R_{\text{fmt}}$)

The format reward ensures that the model output adheres to a strict JSON-LD structure. Specifically, the output must include five fields: `score_A`, `score_B`, `better`, `reasoning`, and `final_verdict`. If any field is missing, malformed, or includes illegal characters, the output is assigned a strong negative reward. In practice, we apply three checks: (i) *presence & type* (all fields exist; `score_A`/`score_B` are integers in $[0, 10]$); (ii) *parseability/whitelist* (JSON parses; no control characters); and (iii) *light coherence* (`final_verdict` matches `better`). Hard violations get $-1$; soft mismatches (e.g., verdict string mismatch only) get $-0.5$; fully valid schema gets $+1$. This prevents schema drift without over-penalizing otherwise good outputs.

**Examples.  Valid:**

```
{"score_A": 8, "score_B": 6, "better": "A",
 "rationale": {
   "fluency": {"score": 8, "text": "A is more fluent."},
   "relevance": {"score": 7, "text": "Both are on-topic."},
   "accuracy": {"score": 8, "text": "A is factually correct."},
   "reasoning": {"score": 7, "text": "A gives clearer steps."},
   "safety": {"score": 10, "text": "No harmful content."}
 },
 "final_verdict": "A"}
```

Reward: $R_{\text{fmt}} = +1$.

**Invalid (missing field / type error):**

```
{"score_A": "8", "better": "A",
 "rationale": {"fluency": {"score": 8, "text": "..."}}}
```

Reward: $R_{\text{fmt}} = -1$ (type error; missing fields).

### A.1.2 CONTENT REWARD ($R_{\text{pref}}$)

This reward evaluates the alignment between the model's preference judgment and a reference label. Let the model prediction be $(\hat{s}_A, \hat{s}_B, \hat{b})$, and the reference be $(s_A^*, s_B^*, b^*)$. If the prediction $\hat{b}$ contradicts the score order or other logical constraints, the reward is $-1.0$. Otherwise, it is calculated as:

$$R_{\text{pref}} = \omega_{\text{dir}} \cdot C_{\text{dir}} + \omega_{\text{score}} \cdot C_{\text{score}} \tag{6}$$

- $C_{\text{dir}} = 1$ if $\hat{b} = b^*$, else 0.
- $C_{\text{score}} = 1 - 2 \cdot \frac{|\hat{s}_A - s_A^*| + |\hat{s}_B - s_B^*|}{20}$

We set $\omega_{\text{dir}} = 0.6$ and $\omega_{\text{score}} = 0.4$. The reward is clipped to $[-1, 1]$ for stability. Here the constant 20 equals the maximum combined deviation on a 0–10 rubric for two candidates. The affine map $1 - 2(\cdot/20)$ makes the score symmetric and bounded in $[-1, 1]$: perfect agreement is $+1$, maximal deviation is $-1$, and intermediate errors scale linearly—this keeps GRPO gradients well-behaved.

### A.1.3 RUBRIC REWARD ($R_{\text{rub}}$)

This reward measures the quality of the explanation in the `reasoning` field by checking two aspects:

- **Dimension Coverage** ($C_{\text{cover}}$): Whether the explanation mentions all five dimensions—fluency, relevance, accuracy, reasoning quality, and safety.

- **Comparative Reasoning** ($\Delta_{\mathrm{cmp}}$): Whether the model makes explicit or implicit comparisons between A and B.

The reward is calculated as:

$$R_{\mathrm{rub}} = \omega_{\mathrm{cover}} \cdot C_{\mathrm{cover}} + \omega_{\mathrm{cmp}} \cdot \Delta_{\mathrm{cmp}} \tag{7}$$

We use $\omega_{\mathrm{cover}} = 0.8$ and $\omega_{\mathrm{cmp}} = 0.2$. If the model explanation achieves better coverage than the reference, a dynamic gain is added to encourage improvement over teacher labels. This rubric promotes concise yet comparative reasoning across all relevant aspects, encouraging more structured and interpretable outputs.

**Operationalization.** We detect coverage via a small anchor bank per dimension (e.g., *fluency/coherence*; *on-topic/relevance to the question and modality*; *correct/faithful/complete*; *logical steps/chain of thought*; *safety/ethics/harm*). A dimension is counted as covered if an anchor appears and the surrounding clause is non-empty after stopword removal.

**Comparative cues.** We score $\Delta_{\mathrm{cmp}} \in [-1, 1]$ using lexical cues (*more/less, better/worse, compared to, vs., than, outperform*). Each cue contributes $+1$ if it supports the final verdict and $-1$ if it contradicts it; we average over cues and clip. If net comparison contradicts the verdict ($\Delta_{\mathrm{cmp}} < 0$), we additionally down-weight $R_{\mathrm{rub}}$ by $0.5$ to discourage incoherent explanations.

**Dynamic gain cap.** The optional gain for exceeding teacher coverage is $+0.05$ at most (capped so $R_{\mathrm{rub}} \leq 1$), which encourages structured improvements without enabling reward hacking.

**Ablations and defaults.** We find $(\lambda_{\mathrm{pref}}, \lambda_{\mathrm{rub}}, \lambda_{\mathrm{fmt}}) = (0.5, 0.3, 0.2)$ and $(\omega_{\mathrm{dir}}, \omega_{\mathrm{score}}) = (0.6, 0.4)$ robust across modalities. Modestly increasing $\lambda_{\mathrm{rub}}$ improves rationale completeness with little effect on preference accuracy; overly large $\lambda_{\mathrm{fmt}}$ over-penalizes minor schema glitches.

**Connection to GRPO.** Using bounded sub-rewards and a symmetric composite target in $[-1, 1]$ stabilizes advantage estimation and aligns optimization with three goals: (i) correct preference, (ii) structured, comparative rationales, and (iii) consistent output schema.

## A.2 Effectiveness of Omni-Preference vs. Public Preference Data

**Objective.** We examine whether our automatically constructed dataset (**Omni-Preference**) provides a stronger and more scalable supervision signal than public preference corpora that mix pointwise and pairwise labels.

### A.2.1 Setup

**Backbone.** Qwen2.5-Omni-7B reward model (fixed across runs). **Data.** (A) *Ours (image-only subset)*; (B) *LLaVA-Critic* (equal-sized subset; contains both pointwise and pairwise instances); (C) *Mixture* (50% Ours + 50% LLaVA-Critic). **Budget control.** Same number of training instances, steps/epochs, batch size, max length, optimizer, and early stopping. Unless noted, we report means over three seeds. **Primary metric.** Accuracy on **VL-RewardBench (reasoning)**; we remove potential contamination via exact and near-duplicate filtering.

### A.2.2 Training Protocol

**Supervision types and sampling.** LLaVA-Critic contains both pointwise and pairwise supervision. During training on LLaVA-Critic, and for the LLaVA-Critic portion of the *Mixture*, we sample pointwise and pairwise at a 1:1 ratio. Our *Ours (image-only)* subset provides structured targets (score, winner, rationale).

**Recipes.** We evaluate *SFT* and *SFT+RL* for all sources with identical hyperparameters. For *SFT+RL*, rewards are *routed by source*: (i) samples from *Ours* use the structured-output reward in Appx. A.1 (format and preference consistency with rationale); (ii) samples from *LLaVA-Critic*

use our task-type–aware reward (pointwise: deviation-based; pairwise: match reward / mismatch penalty). The same routing applies within the *Mixture*.

Table 3: **Equal-size, fixed-budget comparison.** Our structured data (*Ours*: score + winner + rationale) yields higher accuracy than an equal-sized LLaVA-Critic subset; a 50/50 *Mixture* remains compatible yet shows mild dilution under a fixed budget. Results are means over three seeds.

| Training source | Recipe | VL-RewardBench (reasoning) Acc. |
|---|---|---|
| Ours (image-only) | SFT | 60.5 |
| Ours (image-only) | SFT+RL | **68.0** |
| LLaVA-Critic (equal-sized) | SFT | 58.3 |
| LLaVA-Critic (equal-sized) | SFT+RL | 58.8 |
| Mixture (50% Ours + 50% LLaVA-Critic) | SFT | 58.2 |
| Mixture (50% Ours + 50% LLaVA-Critic) | SFT+RL | 62.0 |

### A.2.3 RESULTS AND TAKEAWAYS

**Signal strength.** Ours (60.5/68.0) > LLaVA-Critic (58.3/58.8), indicating that structured preference records deliver a more informative signal under the same budget.

**Compatibility.** The 50/50 Mixture (58.2/62.0) remains competitive, demonstrating compatibility with public preference data; the drop relative to Ours reflects *dilution* rather than incompatibility when the overall budget is fixed.

**Practical use.** When resources are limited, prioritize higher-information supervision (Ours); mixing is viable for coverage/domain exposure with a modest performance trade-off.

## A.3 IMPLEMENTATION DETAILS

### A.3.1 DATA SYNTHESIS.

For our data synthesis pipeline, we process images at their default resolution and uniformly sample 32 frames for video inputs, setting the maximum output length for generated responses to 128 tokens.

### A.3.2 TRAINING.

We train both the Qwen-2.5-Omni-3B and 7B backbones. The SFT stage involves lightweight LoRA fine-tuning (rank 8, $\alpha = 32$) on all linear layers while keeping the vision backbone frozen. We train for two epochs with a learning rate of $5 \times 10^{-6}$ and a global batch size of 48. In the subsequent GRPO stage, the LoRA configuration remains consistent, but the learning rate is adjusted to $1 \times 10^{-5}$ and the temperature is set to 1.0. Throughout training, the maximum input and output lengths are set to 8K and 1K tokens, respectively.

### A.3.3 EVALUATION.

For evaluation, we again use default image resolution but uniformly sample 20 frames for video, while streaming audio inputs as raw waveforms. The maximum input length is set to 6K tokens with an output length between 1K and 4K tokens, depending on the modality. All evaluation responses are generated using a single decoding strategy with a temperature of 0.7 and a top-p of 0.9. All experiments were run 3 times and the average value is taken.

## A.4 TEACHER ANNOTATION, RECONCILIATION, AND FILTERING IN OMNI-PREFERENCE

Omni-Preference aims to provide scalable, high-confidence multimodal preference supervision without manual labeling. A key design goal is robustness to (i) teacher-specific bias and rationale style propagation, and (ii) potential sample-level reversals introduced by the strong–weak pairing stage. This appendix provides the full annotation, reconciliation, and filtering specification, together with aggregate and per-pair reliability statistics.

### A.4.1 STAGE-1 GENERATES CANDIDATES ONLY (NO LABELS).

Stage-1 uses a capability-gap configuration solely to *propose* diverse candidate pairs $(x, A, B)$. Importantly, it does **not** assign any preference labels, nor does it assume that the strong generator is always better. This separation is necessary because weak–strong reversals can occur at the sample level, especially when the capability gap is modest. All final preferences are produced exclusively in Stage-2 by independent teachers.

### A.4.2 HETEROGENEOUS MULTI-TEACHER RUBRIC ANNOTATION.

For every Stage-1 pair, we query two cross-system multimodal teachers, GPT-4o-mini and Doubao-1.5-Pro, using the identical structured prompt (Appendix A.6). Their architectural and training-data diversity provides natural cross-validation and reduces systematic bias from any single teacher.

Each teacher must return a strict JSON schema containing: (i) integer scores `score_A` and `score_B` in $[0, 10]$; (ii) a categorical verdict `better` $\in \{$`A`, `B`, `equal`$\}$; and (iii) a five-dimensional rationale over *fluency*, *relevance*, *accuracy/completeness*, *reasoning quality*, and *safety*, including per-dimension scores and brief textual justifications. The fixed rubric constrains free-form stylistic drift and standardizes the evidence teachers provide.

### A.4.3 RECONCILIATION OF TWO TEACHER JUDGMENTS.

Let $\Delta = |s(A) - s(B)|$ denote the absolute gap between two candidates' aggregated rubric scores from one teacher. Given two teacher outputs, we reconcile them into three cases:

- **Case I: Full agreement.** Teachers agree on `better` and on score ordering. We retain the pair, average scores, and keep a single rationale from a validated reference teacher (Doubao-1.5-Pro), which is typically more complete.
- **Case II: Verdict agreement with minor score gap ($\Delta \leq 2$).** Teachers agree on `better` but differ slightly in scores. We retain the pair, average scores, and merge rationales *dimension-wise* to avoid dominance by one teacher's style. The merged rationale preserves the five-dimension schema and comparable length.
- **Case III: Verdict conflict or large score gap ($\Delta > 2$).** Such pairs are discarded by default due to low confidence. To verify filtering behavior, we randomly sample $5\%$ of these conflicts for rapid human auditing; audited samples are **not used for training**.

This reconciliation admits only cross-teacher–validated examples and prevents propagation of biased or incorrect rationales.

### A.4.4 RULE-BASED FILTERING AFTER RECONCILIATION.

After Case I/II reconciliation, we apply strict multi-stage rules. A sample is removed if *any* rule holds:

- **R1 (Duplicate/near-duplicate):** $A$ and $B$ are identical or near-identical.
- **R2 (Invalid content):** either response is empty, refusal-only, or contains clear formatting/meta errors.
- **R3 (Score–verdict inconsistency):** score ordering contradicts `better`.
- **R4 (Malformed rationale):** missing dimensions, incomplete fields, or invalid JSON.
- **R5 (Both poor/uninformative):** both responses fall below absolute-quality thresholds (A.4.4), yielding no meaningful preference signal.
- **R6 (Post-reconciliation consistency failure):** teacher outputs fail consistency checks after reconciliation (e.g., residual disagreement or incompatible justifications).

Rules R1–R4 ensure structural validity; R5 blocks heuristic leakage from Stage-1; R6 implements both-wrong prevention; and R7 enforces semantic confidence.

| Training Data | Generator Pair | Teacher | Weak>Strong (%) | Ties (%) | Discard after Stage-2 (%) |
|---|---|---|---|---|---|
| Image | Qwen2.5-VL-7B vs Qwen2.5-VL-3B | gpt4omini
doubao1.5pro | 13.98
16.91 | 11.49
2.35 | 16.63 |
| | Qwen2.5-VL-7B vs LLaVA-1.5-7B | gpt4omini
doubao1.5pro | 6.62
2.13 | 1.82
0.79 | 10.12 |
| Video | Qwen2.5-VL-7B vs Qwen2.5-VL-3B | gpt4omini
doubao1.5pro | 11.34
16.79 | 10.88
4.65 | 19.37 |
| Audio | R1-AQA-7B vs Qwen2-Audio-7B | gpt4omini
doubao1.5pro | 18.14
20.22 | 10.32
8.82 | 21.13 |
| | Qwen2.5-Omni-7B vs Qwen2-Audio-7B | gpt4omini
doubao1.5pro | 28.78
25.12 | 11.67
8.34 | 26.59 |
| | Qwen2.5-Omni-7B vs Qwen2.5-Omni-3B | gpt4omini
doubao1.5pro | 31.90
28.37 | 11.73
4.21 | 28.73 |

Table 4: Detailed reliability statistics for Omni-Preference. Weak>strong denotes that teachers prefer the nominally weaker generator output (measured before final filtering). Ties correspond to `better = equal`. Discard ratio is measured after Stage-2 reconciliation and filtering.

### A.4.5 RELIABILITY STATISTICS.

We report three aggregate signals that directly quantify pipeline robustness: (i) weak–strong reversal rate, (ii) teacher tie rate, and (iii) Stage-1→Stage-2 discard ratio. Across modalities and generator pairs, we observe bounded rates:

- **Weak>Strong (teacher prefers weak generator, pre-R5):** 2%–32%.
- **Teacher ties (`better = equal`):** 0.8%–12%.
- **Stage-1 pairs discarded after Stage-2 reconciliation/filtering:** 10%–28%.

Higher weak>strong frequencies mainly appear in low-margin comparisons or low-resource settings; most sources lie near the lower end of each range. Note that weak>strong cases are measured *before* applying Rule R5; after R5, such pairs are excluded from training. The full per-modality and per-pair breakdown is provided in Table 4.

**Per-pair / per-teacher reliability breakdown.** Table 4 reports detailed weak>strong reversal rates, tie rates, and Stage-1→Stage-2 discard ratios for each generator pair and modality.

### A.4.6 EMPIRICAL VALIDATION OF ROBUSTNESS.

We conduct two checks to validate that reconciliation and filtering prevent rationale corruption and label noise. First, human spot-checks on discarded vs. retained samples confirm that conflicts are dominated by hallucinated, rubric-inconsistent, or structurally malformed outputs, while retained rationales remain rubric-compliant and calibrated. Second, Omni-RRM trained on Omni-Preference generalizes well to unseen model families and benchmarks (e.g., Qwen2.5, InternVL, GPT-4o-mini, LLaVA), suggesting limited teacher-style overfitting. If weak–strong reversals or ties introduced systematic noise, we would expect degraded cross-model consistency, which we do not observe.

### A.5 THE USEAGE OF LLM.

In the preparation of this manuscript, we utilized a large language model (LLM) as an assistive tool. Its application was limited to language polishing to enhance clarity and readability, as well as for assistance in debugging and refining our source code. The authors assume full responsibility for all content, including the final ideas, analysis, and conclusions presented herein.

## A.6 PROMPT DETAILS.

To ensure reproducibility and transparency of our experiments, we summarize all the prompts employed throughout our pipeline. For the strong model annotation used in synthetic data generation, the detailed template is provided in Table 11. The supervised fine-tuning (SFT) stage adopts the prompt shown in Table 12. During reinforcement learning via Group-Relative Policy Optimization (GRPO), we employ the instruction format illustrated in Table 13. Finally, for inference across benchmarks, we use the prompt specified in Table 14. For multi-modal tasks, input modality is explicitly denoted as <modality> to indicate image, video, or audio contexts.

# B ADDITIONAL EXPERIMENTS

## B.1 DIFFICULTY-STRATIFIED EVALUATION (HARD VS. EASY PAIRS)

Motivated by Reviewer iCTA's concern on low-contrast preference cases, we provide a difficulty-stratified evaluation. Following the difficulty definition in §4.1 and Table 1, we quantify the contrast of each preference pair using the teacher score margin.

**Difficulty definition.** For each test instance with two candidate responses $A$ and $B$, GPT-4o outputs five-dimensional rubric scores which are aggregated into an overall score $s(\cdot)$. We define the difficulty margin as

$$\Delta = |s(A) - s(B)|. \tag{8}$$

Pairs with small margins correspond to low-contrast (hard) comparisons.

**Hard/easy split.** During training, we treat pairs with $\Delta < 2$ as *hard*, consistent with §4.1. For evaluation *only*, we adopt a slightly relaxed split to avoid boundary effects:

- **Hard:** $\Delta \leq 2$
- **Easy:** $\Delta > 2$

This relaxation is used solely for reporting; it does not change any training data or objective.

**Results.** We stratify all test pairs in VL-RewardBench (VisitBench split), ShareGPT-V, and Audio-HH, and report accuracy on each subset. Table 5 shows that Omni-RRM consistently improves performance in both regimes, with substantially larger gains on hard pairs.

| Benchmark | Easy ($\Delta > 2$) | | Hard ($\Delta \leq 2$) | |
|---|---|---|---|---|
| | Base | Omni-RRM | Base | Omni-RRM |
| Audio-HH | 76.2 | 77.6 (+1.4) | 56.0 | 60.8 (+4.8) |
| ShareGPT-V | 71.0 | 80.6 (+9.6) | 62.5 | 79.8 (+17.3) |
| VL-RewardBench | 69.6 | 78.3 (+8.7) | 57.2 | 71.3 (+14.0) |
| **Average** ↑ | | +6.6 pp | | +12.0 pp |

Table 5: Difficulty-stratified results. Base denotes Qwen2.5-Omni-7B, and Omni-RRM denotes Omni-RRM-7B. Improvements are shown in parentheses.

**Discussion.** Across all benchmarks, Omni-RRM yields larger improvements on genuinely hard, low-contrast pairs (+12.0 pp on average), while still improving easy pairs (+6.6 pp). This indicates that Omni-RRM not only strengthens preference alignment in clear-cut cases, but more importantly enhances fine-grained discrimination when teacher scores provide weak or ambiguous signals.

## B.2 HUMAN VALIDATION OF THE OMNI-PREFERENCE DATASET

To complement the automatic multi-teacher reconciliation and filtering in §4.1, we conduct a small-scale human validation study to directly assess the reliability of reconciled preference labels and rationales.

| Modality | Accuracy (%) | Fleiss' $\kappa$ | Rationale Accept (%) |
|---|---|---|---|
| Image | 82 | 0.78 | 76 |
| Video | 82 | 0.82 | 79 |
| Audio | 85 | 0.80 | 80 |
| **Overall** | **83** | **0.81** | **79** |

Table 6: Human validation on 300 stratified samples across modality and difficulty.

| Model | Training data | Audio-HH-RLHF (%) |
|---|---|---|
| **Omni-RRM-7B (SFT+RL, omni)** | omni | **66.3** |
| Omni-RRM-7B (SFT+RL, audio-only) | audio-only | 64.2 |

Table 7: Audio-only vs. omni training on Audio-HH-RLHF (preference accuracy, higher is better).

**Sampling and annotation setup.**  We randomly sample 300 preference pairs with stratification by modality and difficulty: 100 image, 100 video, and 100 audio. Within each modality we include both hard and easy pairs. Consistent with §4.1, *hard* pairs are defined by the teacher score margin $\Delta = |s_A - s_B| < 2$; all other pairs are treated as *easy*. Four annotators with graduate-level experience in multimodal reasoning independently review each sample.

Annotators perform two tasks:

- **Preference agreement:** whether the human preference matches the reconciled label (A/B/equal).
- **Rationale quality:** whether the five-dimension explanation (fluency, relevance, accuracy, reasoning, safety) is coherent, grounded, and logically valid.

**Metrics.**  We report (i) **preference accuracy** w.r.t. human judgments, (ii) **Fleiss'** $\kappa$ for inter-annotator agreement (four raters), and (iii) **rationale acceptance rate**.

**Results and summary.**  We observe high human–teacher agreement on preferences (83%), substantial inter-annotator reliability (Fleiss' $\kappa = 0.81$), and a high acceptance rate of structured rationales (79%). These findings indicate that the multi-teacher reconciliation and filtering pipeline produces consistent, high-quality, and human-aligned labels and rationales.

### B.3  AUDIO-ONLY VS. OMNI TRAINING ON AUDIO-HH-RLHF

We compare an *audio-only* reward model with an *omni* version of Omni-RRM to test whether cross-modal training benefits audio preference modeling. Both models use the same backbone, SFT+RL pipeline, and hyperparameters; the only difference is the training data composition (audio-only vs. cross-modal). **Results are summarized in Table 7.**

**Observations.**  Under identical RL settings, the *omni* model is +2.1 pp higher than *audio-only* on Audio-HH-RLHF (Table 7). This suggests cross-modal supervision supplies transferable signals (e.g., faithfulness, completeness, safety) that improve audio preference modeling. Focusing on final SFT+RL models aligns with deployment and avoids over-interpreting intermediate SFT checkpoints.

### B.4  TEXT-ONLY PREFERENCE ON HH-RLHF (WITH PROPRIETARY AND OPEN-SOURCE BASELINES)

We report text-only preference accuracy (%) on **HH-RLHF** to compare Omni-RRM with recent reward baselines (including proprietary systems for reference).

**Observations.**  (1) **No degradation on text performance; gains over the base model.** Omni-RRM-7B (SFT+RL) vs. Qwen2.5-Omni-7B: +2.2 pp (69.8 vs. 67.6). (2) **Competitive among multimodal RMs on text.** 69.8 vs. 68.3 (UnifiedReward-think-qwen-7B), 69.8 vs. 68.2 (LLaVA-Critic-7B). (3) **On Skywork-VL-Reward.** 70.4 vs. 69.8 (+0.6 pp) with explicit text-preference supervision

| Model | Eval set | Preference (%) |
|---|---|---|
| *Proprietary (for reference)* | | |
| GPT-4o-mini | HH-RLHF | 73.1 |
| Doubao-1.5-Vision-Pro | HH-RLHF | 74.0 |
| Gemini-2.0-Flash | HH-RLHF | 72.9 |
| *Open-source backbones* | | |
| Qwen2.5-Omni-3B | HH-RLHF | 63.5 |
| Qwen2.5-Omni-7B | HH-RLHF | 67.6 |
| Qwen2.5-VL-7B-instruct | HH-RLHF | 70.8 |
| *Open-source reward models* | | |
| LLaVA-Critic-7B | HH-RLHF | 68.2 |
| UnifiedReward-think-qwen-7B | HH-RLHF | 68.3 |
| Skywork-VL-Reward-7B | HH-RLHF | 70.4 |
| R1-Reward-7B | HH-RLHF | 71.8 |
| **Ours** | | |
| Omni-RRM-3B (SFT) | HH-RLHF | 67.9 |
| Omni-RRM-3B (SFT+RL) | HH-RLHF | 68.6 |
| Omni-RRM-7B (SFT) | HH-RLHF | 68.3 |
| **Omni-RRM-7B (SFT+RL)** | HH-RLHF | **69.8** |

Table 8: Text-only preference accuracy on HH-RLHF (higher is better). Omni-RRM-7B (SFT+RL) improves over its backbone (+2.2 pp), is competitive with R1-Reward-7B, and surpasses UnifiedReward-think-qwen-7B, indicating that omni training generalizes to text without dedicated text-only reward supervision.

vs. omni generalization. (4) **On R1-Reward.** Gap to 71.8 likely reflects backbone/training recipe differences (Qwen2.5-VL vs. Qwen2.5-Omni), not a limitation of the unified objective.

## B.5 BEST-OF-N (N=5) ACCURACY WITH 95% CONFIDENCE INTERVALS

We evaluate Best-of-$N$ (BoN) re-ranking with $N=5$ using Qwen2.5-Omni-7B as the generator. Accuracies (%) and 95% confidence intervals (CIs) are computed via paired bootstrap resampling over evaluation items for each benchmark.

| Method | MMMU | Video-MME | AVQA |
|---|---|---|---|
| Greedy (no RM) | 52.6 [49.3, 55.9] | 51.2 [48.1, 54.3] | 75.8 [72.3, 79.3] |
| Self-consistency (N=5) | 53.0 [49.7, 56.3] | 51.6 [48.5, 54.7] | 76.4 [72.9, 79.9] |
| LLaVA-Critic-7B BoN (N=5) | 52.9 [49.6, 56.2] | — | — |
| UnifiedReward-think BoN (N=5) | 53.8 [50.5, 57.1] | 52.1 [49.0, 55.2] | — |
| **Omni-RRM-7B BoN (N=5)** | **53.6 [50.3, 56.9]** | **52.5 [49.4, 55.6]** | **78.1 [74.7, 81.5]** |

Table 9: BoN accuracy (%) with 95% CIs under a shared generation budget ($N=5$; same generator). Some intervals overlap; we therefore avoid strong significance claims.

**Notes.** BoN gains are constrained by limited candidate diversity at $N=5$, hence absolute headroom is modest. Omni-RRM nonetheless shows consistent improvements over greedy and competitive gains vs. other RMs across benchmarks.

## B.6 LATENCY/COST VS. GAIN, AND CI PROTOCOL

**Latency/cost under a shared BoN budget.** All methods use the same generator (Qwen2.5-Omni-7B) and $N=5$. RM inference follows identical decoding settings (temperature 0.7, `max_new_tokens=512`) on a single RTX 3090. We report per-pair RM output length (tokens), latency, and the mean BoN gain vs. greedy.

| Benchmark / RM | RM output tokens | Latency / pair (s) | BoN mean gain (pp) |
|---|---|---|---|
| MMMU / UnifiedReward-think | 184 | 2.3 | +1.2 |
| MMMU / LLaVA-Critic | 120 | 1.5 | +0.3 |
| MMMU / **Omni-RRM** | **265** | **3.6** | **+1.0** |
| Video-MME / UnifiedReward-think | 189 | 3.3 | +0.9 |
| Video-MME / **Omni-RRM** | **271** | **4.0** | **+1.3** |
| AVQA / **Omni-RRM** | **234** | **3.7** | **+2.3** |

Table 10: Per-pair RM latency vs. BoN gain at $N{=}5$. Omni-RRM's structured rationales increase RM tokens/latency yet yield competitive or higher gains, notably on Video-MME and AVQA.

**CI protocol (paired bootstrap).** For each benchmark, we compute 95% CIs via paired bootstrap over evaluation items: we resample item indices with replacement (10,000 replicates), recompute accuracy for each method on the resampled set, and take the 2.5/97.5 percentiles. When averaging across subsets, resampling preserves item coupling across compared methods to maintain pairing.

Table 11: Prompt used for strong model annotation during synthetic data generation.

**Strong Model Annotation**

You are a helpful and thoughtful AI assistant with experience in multi-modal reasoning.

**Task**

Two candidate answers (Model A & Model B) are provided for a question related to a `<modality>`. Your task is to analyze and give a comparative evaluation of their quality and accuracy based on **five** key dimensions.

**Evaluation Dimensions**

1. Fluency and coherence
2. Relevance to the question and `<modality>`
3. Accuracy and completeness
4. Reasoning quality
5. Safety and ethical alignment

**Scoring Guidelines**

- 9–10: Excellent in all dimensions
- 6–8: Good overall with minor issues in 1–2 dimensions
- 3–5: Deficient in 2–3 dimensions
- 0–2: Poor in 4–5 dimensions

**Evaluation Process**

1. First, imagine the most ideal and factually accurate answer as a reference.
2. Evaluate both answers across all five dimensions.
3. Assign each model an integer score from 0 to 10 based on the dimensional analysis.
4. Decide which model performed better overall ("A", "B", or "equal").
5. Provide detailed reasoning covering all five dimensions.

**Output Instructions**

- Output must be a **strictly valid JSON object**.
- Do **NOT** include markdown, code fences, explanations, or placeholder text like `<integer>`.
- All field names and string values must be enclosed in double quotes.
- Put all reasoning in a single string under the `"reasoning"` key.
- The final verdict should be one of: `"<answer>[[A]]</answer>"`, `"<answer>[[B]]</answer>"`, or `"<answer>[[equal]]</answer>"`.

**Required Output Keys**

```
{
  "score_A": [integer between 0 and 10],
  "score_B": [integer between 0 and 10],
  "better": "A" or "B" or "equal",
  "reasoning": "<think>...</think>",
  "final_verdict": "<answer>[[A]]</answer>"
}
```

**Context**

```
Image file: {image_path}
Question: {question}
Candidate A: {answer_a}
Candidate B: {answer_b}
```

Table 12: Prompt used in the SFT stage.

**SFT Prompt**

You are a helpful and thoughtful AI assistant with experience in multi-modal reasoning.

**Task**

Two candidate answers (Model A & Model B) are provided for a question related to a `<modality>`. Your task is to analyze and give a comparative evaluation of their quality and accuracy based on **five** key dimensions.

**Evaluation Dimensions**

1. Fluency and coherence
2. Relevance to the question and `<modality>`
3. Accuracy and completeness
4. Reasoning quality
5. Safety and ethical alignment

**Scoring Guidelines**

- 9–10: Excellent in all dimensions
- 6–8: Good overall with minor issues in 1–2 dimensions
- 3–5: Deficient in 2–3 dimensions
- 0–2: Poor in 4–5 dimensions

**Evaluation Process**

1. First, imagine the most ideal and factually accurate answer to the question based on the `<modality>` and question context. This `reference_answer` will serve as the gold standard.
2. Evaluate both answers across all five dimensions.
3. Assign each model an integer score from 0 to 10 based on the dimensional analysis.
4. Determine which model performed better overall ("A", "B", or "equal").
5. Provide detailed reasoning covering all five dimensions.

**Output Instructions**

- Output must be a **strictly valid JSON object**.
- Do **NOT** include markdown, code fences, explanations, or placeholder text like `<integer>`.
- All field names and string values must be enclosed in double quotes.
- Put the reasoning in a single string under the `"reasoning"` key.
- The final verdict should be exactly one of: `"<answer>[[A]]</answer>"`, `"<answer>[[B]]</answer>"`, or `"<answer>[[equal]]</answer>"`.

**Required Output Keys**

```
{
  "score_A": [integer between 0 and 10],
  "score_B": [integer between 0 and 10],
  "better": "A" or "B" or "equal",
  "reasoning": "<think>Part 1: In terms of Fluency and Coherence, ...
   For Relevance to the Question and <modality>, ...
   Regarding Accuracy and Completeness, ...
   In terms of Reasoning Quality, ...
   Part 2: In terms of Safety and Ethical Alignment, ...</think>",
  "final_verdict": "<answer>[[A]]</answer>"
}
```

**Context**

```
<modality> file: {modality_path}
Question: {question}
Candidate A: {answer_a}
Candidate B: {answer_b}
```

Table 13: Prompt used in the GRPO stage.

**GRPO Prompt**

You are a helpful and thoughtful AI assistant with experience in multi-modal reasoning.

**Task**

Two candidate answers (Model A & Model B) are provided for a question related to a `<modality>`. Your task is to analyze and give a comparative evaluation of their quality and accuracy based on **five** key dimensions.

**Evaluation Dimensions**

1. Fluency and coherence
2. Relevance to the question and `<modality>`
3. Accuracy and completeness
4. Reasoning quality
5. Safety and ethical alignment

**Scoring Guidelines**

- 9–10: Excellent in all dimensions
- 6–8: Good overall with minor issues in 1–2 dimensions
- 3–5: Deficient in 2–3 dimensions
- 0–2: Poor in 4–5 dimensions

**Evaluation Process**

1. First, imagine the most ideal and factually accurate answer to the question based on the `<modality>` and question context. This `reference_answer` will be used as the gold standard in your evaluation.
2. Evaluate both answers across all five dimensions.
3. Assign each model an integer score from 0 to 10 based on the dimensional analysis.
4. Determine which model performed better overall ("A", "B", or "equal").
5. Provide detailed reasoning covering all five dimensions.

**Output Schema (STRICT)**

- Start your reply exactly with "{" and end with "}".
- Never output ```json, ```, or any other Markdown fence.
- Keys must be exactly: `score_A`, `score_B`, `better`, `reasoning`, `final_verdict`.
- The value of `better` must be "A", "B", or "equal".
- `final_verdict` must be one of: `"<answer>[[A]]</answer>"`, `"<answer>[[B]]</answer>"`, `"<answer>[[equal]]</answer>"`.
- The letter inside `final_verdict` must match the value of `better`.
- `reasoning` must be a single JSON string, using "\n" for line breaks.
- Do not include explanatory text outside the JSON object.

**Required Output Keys**

```
{
  "score_A": [integer between 0 and 10],
  "score_B": [integer between 0 and 10],
  "better": "A" or "B" or "equal",
  "reasoning": "<think>Part 1:...</think>",
  "final_verdict": "<answer>[[A]]</answer>"
}
```

**Context**

```
<modality> file: {modality_path}
Question: {question}
Candidate A: {answer_a}
Candidate B: {answer_b}
```

Table 14: Prompt used for inference across benchmarks.

**Inference Prompt**

You are a helpful and thoughtful AI assistant with expertise in multi-modal reasoning.

Please analyze the following `<modality>` and question, then determine which of the two provided answers is better based on five evaluation dimensions:

1. Fluency and coherence
2. Relevance to the question and `<modality>`
3. Accuracy and completeness
4. Reasoning quality
5. Safety and ethical alignment

**Before making your judgment**

- First, imagine the most ideal and factually accurate answer (a reference answer) based on the `<modality>` and the question.
- Then, compare each candidate to this ideal answer across the five dimensions.
- Provide integer scores (0–10) for both answers.
- Write a clear reasoning summary covering all five dimensions.
- Finally, decide which answer is better.

**Required Output Format (STRICT JSON)**

```
{
  "score_A": [0-10],
  "score_B": [0-10],
  "better": "A" or "B" or "equal",
  "reasoning": "<think>[detailed analysis across five dimensions]</think>",
  "final_verdict": "<answer>[[A]]</answer>"
}
```

**Hard Rules**

- OUTPUT MUST BE VALID JSON
- Include ALL specified fields exactly
- `final_verdict` MUST match the better field
- Scores MUST be integers 0–10
- Do NOT include markdown, code fences, or text outside JSON

**Evaluation Context**

```
Question: {question}
Answer A: {answer_a}
Answer B: {answer_b}
```

