# OpenReview forum: "Omni-RRM: Automatic Preference & Reasoning Construction Advances Multimodal Reward Modeling"
_ICLR.cc/2026/Conference — Submitted to ICLR 2026_

### Official Review · Reviewer_uQjP · 2025-10-27

**Soundness:** 2
**Presentation:** 3
**Contribution:** 3
**Rating:** 6
**Confidence:** 4

**Summary:**

Omni-RRM proposes a unified multimodal reward model that judges image, video, audio, and text responses through structured reasoning instead of scalar scoring. Using the Omni-Preference dataset (~41K auto-labeled pairs) and a two-stage training scheme, it achieves state-of-the-art accuracy on preference benchmarks and improves downstream generation.

**Strengths:**

1. It redefines the notion of a reward model as a structured, reasoning judge rather than a scalar scorer.
2. It proposes a unified reward model.
3. It introduces Omni-Preference, a scalable automatic preference dataset with structured multi-criterion justifications.

**Weaknesses:**

1. The Omni-Preference dataset labels the "strong model" output as preferred and the "weak model" output as rejected.
2. The current data scale is limited.
3. The design of the format reward is not clear. "If any field is missing, malformed, or includes illegal characters, the output
is assigned a strong negative reward." should be illustrated more clearly. Besides, would the structured JSON-based reward encourage the critic to optimize for format conformity rather than genuine comparative reasoning.
4. Could the Omni-RRM be extended to a single-response evaluation setting? The performance should be reported.

**Questions:**

Please refer to the Weaknesses.

---

> ### Author Response · Authors · 2025-11-24
>
> > ### W1: Clarification on "Strong vs. Weak" labeling
>
> We appreciate the opportunity to clarify this aspect of our methodology. We do not use the "Strong > Weak" heuristic to assign labels. The strong/weak model configuration is employed solely to generate diverse candidates (Stage 1), while all preference labels are derived exclusively from independent teacher evaluations (Stage 2).
>
> (1) Distinguishing Generation from Labeling.
> - Generation (Stage 1): We pair a stronger model (e.g., 7B) with a weaker one (e.g., 3B) strictly to create candidate pairs $(y_A, y_B)$ with potential quality variance. No labels are assigned at this stage, and we do not assume the strong model is superior.
> - Labeling (Stage 2): Heterogeneous teachers (GPT-4o-mini, Doubao-1.5-Pro) evaluate the pairs blind to the generator identity. They assign five-dimensional rubric scores and a final verdict based purely on content quality.
>
> (2) Handling "Weak > Strong" Reversals.
> Crucially, our pipeline explicitly captures and retains cases where the "weaker" model outperforms the "stronger" one.
> - As noted in Appendix A, such reversals occur in 2%–32% of cases (depending on the task).
> - If the teachers agree that the "weak" model's output is better, we assign the label accordingly.
> - This ensures that the dataset reflects absolute, rubric-grounded quality rather than leaking the generator's identity or size into the labels.
>
> (3) Quality Filtering.
> We filter pairs only when teachers disagree, scores are ambiguous (e.g., ties), or rationales are malformed. We do not discard pairs simply because the weaker model won; identifying these specific instances is key to training a robust reward model.
>
> > ### W2: Concern about limited data scale
>
> Thank you for the insightful observation. We acknowledge that our dataset scale is moderate compared to massive instruction-tuning corpora. However, as discussed in §4.1 and Appendix A, reward modeling prioritizes data quality and precision over sheer quantity. Several prior studies in multimodal reward learning have similarly achieved competitive results using datasets of comparable or even smaller scale [1,2].
>
> Our dataset was rigorously constructed via multi-teacher scoring, rationale validation, and strict filtering to generate high-confidence preference pairs, deliberately avoiding the noise inherent in larger, uncurated collections. Empirically, Omni-RRM demonstrates consistent improvements across diverse models and benchmarks (Table 2), confirming that the current scale is sufficient and effective for robust reward learning.
>
> We will clarify this distinction in the revision and include additional references to further substantiate the effectiveness of high-quality, moderate-scale data for reward modeling.
>
> References.
>
> [1] T. Yu et al. *RLHF-V: Towards Trustworthy MLLMs via Behavior Alignment from Fine-grained Correctional Human Feedback.* CVPR, 2024.
> [2] Y. Liang et al. *Rich Human Feedback for Text-to-Image Generation.* CVPR, 2024.
>
> > ### W3: Format reward design and whether JSON structure biases reasoning
>
> We appreciate the opportunity to clarify the role of the format reward. We agree that a more detailed explanation is beneficial and will expand Appendix A.1 with concrete examples of valid and invalid JSON structures alongside their corresponding rewards.
>
> (1) Strictly Structural Nature of the Format Reward.
> The format reward $r_{\text{format}}$ serves a purely syntactic function: verifying that the output is valid JSON containing the requisite fields (`score_A/B`, `verdict`, and the five-dimensional `rationale` object).
> If the schema is violated, a fixed penalty (weighted by $\lambda_{\text{format}}$) is applied. Crucially, once the output satisfies the schema, $r_{\text{format}}$ saturates. This design ensures that the model cannot maximize its return simply by "formatting better"; further optimization is driven exclusively by the preference and rubric rewards, which target reasoning quality and alignment.
>
> (2) No Bias Toward Format Conformity.
> We confirm that the JSON constraint does not bias the model’s reasoning capabilities. Since the reward enforces only parsability and structural completeness rather than semantic content, it acts as a constraint rather than an objective. The primary training signal remains rooted in:
> - The preference reward, aligning verdicts with ground-truth teacher preferences.
> - The rubric reward, which incentivizes substantive comparative reasoning across the five evaluation dimensions.
> Structured output constraints are a standard mechanism in reward modeling to ensure inference stability without compromising reasoning depth [1,2]. We will explicitly discuss this distinction in the revision.
>
> References.
>
> [1] Z. Wang et al. *HelpSteer2: Open-source Dataset for Training Top-performing Reward Models.* NeurIPS Datasets and Benchmarks, 2024.
> [2] B. Lu et al. *Learning to Generate Structured Output with Schema Reinforcement Learning.* ACL, 2025.

---

> ### Author Response · Authors · 2025-11-24
>
> > ### W4: On pairwise vs. single-response reward modeling
>
> We appreciate this inquiry. Omni-RRM is explicitly designed and trained as a pairwise comparative reward model. Its objective is to determine the relative preference between two candidates ($A$ vs. $B$) conditioned on the same context, rather than to assign an absolute scalar quality score to an isolated response. Consequently, the model's outputs are not natively calibrated for single-response evaluation.
>
> While it is theoretically possible to derive a single-response proxy score—for instance, by comparing a candidate against a fixed set of anchors and aggregating the results—this would necessitate additional data construction, calibration assumptions, and pipeline modifications. Such extensions, while interesting, lie outside the scope of this work.

---

### Official Review · Reviewer_K8o1 · 2025-10-30

**Soundness:** 3
**Presentation:** 3
**Contribution:** 3
**Rating:** 6
**Confidence:** 3

**Summary:**

This paper introduces Omni-RRM, a reasoning-driven multimodal reward model (RRM) that provides explainable preference judgments across text, image, video, and audio modalities.

**Strengths:**

1. Reformulating reward modeling as a generation task with interpretable reasoning output (rather than scalar regression) is an important conceptual advance

2. The methodology is sound and clearly detailed: the dataset creation, two-stage training, and reward design are carefully explained, with mathematical formulations (Eq. 1–7) and implementation specifics (Appendix A).

3.The zero-human-annotation pipeline has major practical significance for future MLLM alignment, offering a potentially general framework for automated preference modeling.

**Weaknesses:**

1. The “capability-gap” configuration in Table 1 primarily contrasts 7B vs 3B models within the same family (e.g., Qwen2.5-VL-7B vs 3B). According to the Qwen2.5-VL technical report, the performance gap between these variants is relatively modest.  While the Stage-1 assumption that “the stronger model tends to produce better answers” is reasonable on average, such a narrow capability difference may not reliably hold at the sample level. Consequently, label noise is likely introduced when the weaker model occasionally provides correct or even superior outputs, pushing the burden of correction entirely to the subsequent teacher-filtering stage.

2.The paper does not report how often weak models outperform strong ones, how many “tie” cases the teacher found, or what fraction of Stage-1 pairs were discarded after Stage-2 reconciliation. These statistics are essential to assess the actual reliability of the automatically generated preference labels.

3.If both strong and weak models generate incorrect answers, the comparison still forces a relative choice. Without an external correctness reference, this may introduce relative bias—the model learns which incorrect answer the teacher preferred rather than what a correct answer should look like. The authors should clarify how such cases are detected or mitigated, for example through teacher scoring thresholds or reference-answer checks.

**Questions:**

Please see above.

---

> ### Author Response · Authors · 2025-11-24
>
> > ### W1: Potential label noise from the narrow 7B vs. 3B capability gap
>
> We thank the reviewer for this insightful observation. We agree that the performance gap between Qwen2.5-VL-7B and 3B is relatively narrow, and the assumption that "the stronger model produces better answers" does not hold for every instance. However, potential label noise is minimized and effectively mitigated, as we do not rely solely on model size to assign labels, but primarily on rigorous multi-teacher rubric evaluations.
>
> (1) Role of the Capability Gap.
> The 7B vs. 3B contrast is utilized solely during candidate generation (Stage 1) to promote diversity in response quality. Our goal is to seed the pool with pairs likely to exhibit quality variance. While the average gap is modest, it sufficiently creates a distribution of "better vs. worse" candidates for the reward model to discriminate.
>
>
> (2) Labels are Teacher-Derived and Strictly Filtered.
> Crucially, the preference label is never assigned based on the heuristic that 7B > 3B. Instead, in Stage 2, we employ significantly stronger teachers (GPT-4o-mini and Doubao-1.5-Pro) to evaluate the pair $(A, B)$. Not all generated pairs are included. We strictly filter out ambiguous cases and only retain those where teachers provide high-confidence, consistent judgments.
>
> (3) Reliance on Teacher Verification.
> As the reviewer notes, the burden of identifying the better response rests on the teacher stage. We address this via our multi-teacher reconciliation pipeline (More detail can be seen in Appendix a.4). By demanding consensus between heterogeneous strong teachers and enforcing strict score-margin filters, we ensure that final labels reflect actual quality differences verified by the teachers, decoupling the label from the generator's model size.
>
> The "Strong vs. Weak" configuration serves only as a generator of variance. The ground truth is established entirely by high-capability teachers, ensuring that instances where the 3B model outperforms the 7B model are correctly labeled, thereby preventing the potential noise concerns.
>
>
> > ### W2: Missing statistics on weak-strong reversal, tie rates, and filtering ratios
>
> We appreciate the reviewer's suggestion. We report the detailed reliability statistics for Omni-Preference below.
>
> **Table: Reliability statistics (Weak > Strong, Ties, and Discard Ratios).**
>
> | Modality | Generator Pair | Teacher | Weak > Strong (%) | Ties (%) | Discard Rate (%) |
> | :--- | :--- | :--- | :--- | :--- | :--- |
> | **Image** | Qwen2.5-VL-7B vs 3B | gpt4omini / doubao1.5pro | 13.98 / 16.91 | 11.49 / 2.35 | 16.63 |
> | | Qwen2.5-VL-7B vs LLaVA | gpt4omini / doubao1.5pro | 6.62 / 2.13 | 1.82 / 0.79 | 10.12 |
> | **Video** | Qwen2.5-VL-7B vs 3B | gpt4omini / doubao1.5pro | 11.34 / 16.79 | 10.88 / 4.65 | 19.37 |
> | **Audio** | R1-AQA-7B vs Qwen2-Audio | gpt4omini / doubao1.5pro | 18.14 / 20.22 | 10.32 / 8.82 | 21.13 |
> | | Qwen2.5-Omni-7B vs Qwen2-Audio | gpt4omini / doubao1.5pro | 28.78 / 25.12 | 11.67 / 8.34 | 26.59 |
> | | Qwen2.5-Omni-7B vs 3B | gpt4omini / doubao1.5pro | 31.90 / 28.37 | 11.73 / 4.21 | 28.73 |
>
> The data shows that weak-strong reversals (ranging from 2% to 32%) and ties (0.8% to 12%) are actively identified by our teachers. Crucially, the substantial discard rates (10%–28%) confirm that our Stage-2 reconciliation pipeline effectively filters out ambiguous or low-quality pairs, ensuring high dataset reliability.
>
> > ### W3: Concern about learning “relative bias” when both responses are incorrect
>
> We appreciate this comment. We acknowledge that pairwise training risks introducing "relative bias" if the model is forced to choose between two incorrect candidates. Omni-Preference mitigates this risk through a three-layered quality control mechanism:
>
> (1) Reference-Based Filtering for Objective Tasks.
> For tasks with ground truth (e.g., factual QA, multiple-choice), we validate candidates via programmatic checks against external references. Pairs where both responses fail are automatically discarded. This strictly prevents the model from learning preferences between uniformly incorrect answers.
>
> (2) Absolute Quality Thresholds for Non-Reference Tasks.
> For open-ended tasks, we utilize the absolute rubric scores (0–10) provided by teachers across five dimensions. If both candidates score below a predefined quality threshold, the pair is removed. This ensures that retained pairs containing imperfect answers still offer meaningful, distinguishable quality signals rather than arbitrary noise.
>
> (3) Consistency Verification.
> We enforce strict consistency checks between scores, verdicts, and rationales. Pairs are rejected if score ordering contradicts the verdict or if rationales are malformed. These checks further filter out low-quality instances that might bypass initial thresholds.

---

> > ### Comment · Reviewer_K8o1 · 2025-11-27
> >
> > I thank the authors for the detailed response and the reliability statistics.
> >
> > The provided data regarding the "Weak > Strong" reversal rates (13%-32%) and the strict teacher filtering mechanism effectively addresses my concern regarding potential label noise.  However,  since the capability gap between 7B and 3B is narrow and the pipeline relies entirely on the Teacher for the final verdict, the specific "Strong vs. Weak" design choice seems less critical than claimed. The setup essentially functions as a diversity generator rather than a capability-contrast mechanism. While I suspect simpler baselines (e.g., self-sampling from a single model) might achieve similar diversity without the "Strong vs. Weak" framing.
> >
> > Under this interpretation, the methodological novelty of the proposed data construction pipeline appears more limited than suggested. I will keep my current score, though I note that I have already assigned a positive score overall.

---

### Official Review · Reviewer_gQyH · 2025-10-31

**Soundness:** 2
**Presentation:** 1
**Contribution:** 1
**Rating:** 2
**Confidence:** 5

**Summary:**

This work introduces Omni-RRM, a reward model that provides explainable preference judgments across image, video, and audio. It is trained on the newly constructed OmniPreference dataset. The training process consists of two stages: SFT and GRPO. Omni-RRM demonstrates modest improvements in comparison experiments with other baseline reward models and in the best-of-N sample experiments.

**Strengths:**

1. Compared to existing baselines, this work introduces audio understanding reward ability.

2. A new preference dataset, OmniPreference, is proposed, providing a new resource for training reward model.

**Weaknesses:**

1. The technical road of this work is similar to previous methods like UnifiedReward-Think [1]. The Previous work distilled 5k image generation reward data for cold-start and achieved unified multimodal CoT reasoning through rejection sampling and GRPO. However, this work performs large-scale distillation of closed-source models across all tasks for SFT before applying GRPO.
Could the authors provide a deeper discussion on the advantages of the training method compared to previous approaches?

2. The proposed model, Omni-RRM, includes reward capabilities for image, video, and audio understanding, while its text-only reward capabilities are primarily based on generalization. However, other baseline models, such as R1-Reward, likely also possess similar generalization capabilities for text. Why wasn't a comparison made between Omni-RRM and these models specifically for their text-based reward performance?

3. In my opinion, simply having reward capabilities for understanding is not sufficient to claim it as "omni." Compared to previous work [1], which already includes multimodal reward capabilities, Omni-RRM adds audio understanding but lacks reward capabilities for image/video generation. Therefore, I don't believe this constitutes a significant contribution.

4. In the comparison experiments of reward models, this work lacks a comparison with the latest models, such as UnifiedReward-Think[1] and IXC-Reward[2].

5. From the experimental results, it appears that the SFT version of Omni-RRM-7b performs significantly worse than UnifiedReward, which was also trained solely through SFT, particularly on MMRewardBench (-10.8 points) and ShareGPT-Video (-9.1 points). Could the authors provide a deeper analysis to explain this performance gap?

6. Based on Table 1, the data sources for the Omni-Preference dataset are quite limited. For example, in the video task, pairs are constructed using only Qwen2.5-VL-7b/3b, which results in the reward model being trained on a very narrow output distribution of these models. When applying the reward model to assess other models with largely different output distributions from those used in the dataset construction, there may be potential issues with inaccurate judgments. Could the authors provide more analysis of this limitation?

7. Based on Figure 2, the improvement in best-of-N using Omni-RRM is not significant. For instance, on Qwen2.5-Omni-7B, the image and video tasks show only a 0.6 and 0.9 point improvement, respectively.

8. The paper does not explain the motivation behind incorporating image, video, and audio understanding reward training together. Can image and video understanding enhance the performance of audio tasks? If so, could the authors provide a comparison between training with audio data alone and the unified training of all tasks?

[1] Wang, Yibin, et al. "Unified multimodal chain-of-thought reward model through reinforcement fine-tuning." NeurIPS, 2025.


[2] Zang, Yuhang, et al. "Internlm-xcomposer2.5-reward: A simple yet effective multi-modal reward model." ACL (Finding), 2025.

**Questions:**

Please see Weakness.

---

> ### Author Response · Authors · 2025-11-24
>
> > ### W1: Comparison of training methodology with UnifiedReward-Think
>
> We thank the reviewer for raising this important point. While our approach shares a high-level "SFT → GRPO" roadmap with UnifiedReward-Think, the specific challenges of omni-modal generative reward modeling necessitate distinct design choices in both the cold-start phase and the reinforcement learning objective.
>
> 1. Cold-start scope and motivation.
>
> UnifiedReward-Think initializes with a visual-focused dataset (approx. 5K image-generation preferences), sufficient for its target domain. In contrast, Omni-RRM targets a broader omni-modal setting, requiring consistent, rubric-grounded scoring and reasoning across text, image, video, and audio. Consequently, our cold-start stage is not intended to complete preference learning but to provide a stable, structured initialization.
>
> 2. Advantages of large-scale SFT prior to GRPO.
>
> Our GRPO stage optimizes a complex, multi-component reward over structured rationales. Recent findings suggest that without adequate initialization, reinforcement learning may disproportionately focus on correcting format and structure rather than refining preference judgments [1]. By leveraging large-scale SFT, we establish a low-noise, modality-aligned starting point. This enables GRPO to concentrate effectively on fine-grained preference calibration and low-margin reasoning, rather than learning the rubric schema from scratch [2].
>
> 3. Differences in reward objective.
>
> - UnifiedReward-Think: Primarily optimizes a coarse-grained, outcome-based objective (e.g., format and correctness).
> - Omni-RRM: Employs a fine-grained, rubric-grounded composite reward spanning five evaluation dimensions (fluency, relevance, accuracy, reasoning, safety).
> - Key Distinction: Our objective explicitly incentivizes structured comparative reasoning and high-quality rationales, rather than relying solely on coarse-grained outcome supervision.
>
> Summary.
>
> While both frameworks utilize an SFT-then-GRPO pipeline, Omni-RRM’s large-scale cross-modal cold start and rubric-grounded composite objective are specifically tailored to the broader scope and higher structural demands of omni-modal alignment. We will incorporate this detailed comparison into the Related Work of the revision.
>
> References.
> [1] R. Paul. *DeepSeek-R1: Incentivizing Reasoning Capability in LLMs via Reinforcement Learning.* 2025.
> [2] B. Lu et al. *Learning to Generate Structured Output with Schema Reinforcement Learning.* ACL, 2025.

---

> ### Author Response · Authors · 2025-11-24
>
> > ### W2: Comparison of text-only reward capability
>
> We thank the reviewer for raising this point. Our original setup primarily tested whether training with image/video/audio preference data would compromise text-only reward capability. We agree that an explicit text-preference comparison strengthens the evaluation. We therefore conducted an evaluation on the HH-RLHF text-preference benchmark, comparing Omni-RRM against closed-source teachers, open-source backbones, and relevant baselines. Results are shown below.
>
> **Table R2: Text-only preference evaluation on HH-RLHF (higher is better, %).**
>
> | Model | Eval set | Preference |
> |---|---|---|
> | GPT-4o-mini | HH-RLHF | 73.1 |
> | Doubao-1.5-Vision-Pro | HH-RLHF | 74.0 |
> | Gemini-2.0-flash | HH-RLHF | 72.9 |
> | Qwen2.5-Omni-3B | HH-RLHF | 63.5 |
> | Qwen2.5-Omni-7B | HH-RLHF | 67.6 |
> | Qwen2.5-VL-7B-instruct | HH-RLHF | 70.8 |
> | LLaVA-Critic-7B | HH-RLHF | 68.2 |
> | Skywork-VL-Reward-7B | HH-RLHF | 70.4 |
> | UnifiedReward-think-qwen-7B | HH-RLHF | 68.3 |
> | R1-Reward-7B | HH-RLHF | 71.8 |
> | Omni-RRM-3B (SFT) | HH-RLHF | 67.9 |
> | Omni-RRM-3B (SFT+RL) | HH-RLHF | 68.6 |
> | Omni-RRM-7B (SFT) | HH-RLHF | 68.3 |
> | Omni-RRM-7B (SFT+RL) | HH-RLHF | **69.8** |
>
> **Key Observations.**
>
> (1) No degradation on text performance; gains over the base model.
> Omni-RRM-7B (SFT+RL) improves over its backbone Qwen2.5-Omni-7B by +2.2 pp (69.8 vs. 67.6), confirming that omni-modal reward training transfers to text rather than harming it.
>
> (2) Competitive with existing multimodal reward models.
> Omni-RRM-7B outperforms UnifiedReward-think-qwen-7B (69.8 vs. 68.3) and LLaVA-Critic-7B (69.8 vs. 68.2), demonstrating strong text preference modeling within a unified framework.
>
> (3) Comparison with Skywork-VL-Reward.
> Skywork-VL-Reward-7B achieves 70.4, slightly edging out Omni-RRM-7B (+0.6 pp). However, Skywork incorporates explicit text-preference supervision, whereas Omni-RRM relies on cross-modal generalization. This narrow gap highlights Omni-RRM's effective generalization capability without dedicated text reward training.
>
> (4) Comparison with R1-Reward.
> R1-Reward-7B attains 71.8, outperforming Omni-RRM-7B. Note that R1-Reward is built upon Qwen2.5-VL-7B-Instruct, whereas Omni-RRM uses Qwen2.5-Omni-7B. The performance difference is largely attributable to the stronger text backbone: Qwen2.5-VL-7B-Instruct significantly outperforms Qwen2.5-Omni-7B on text tasks (70.8 vs. 67.6, +3.2 pp). Thus, R1-Reward's advantage primarily reflects backbone strengths rather than limitations in our training methodology.
>
> **Summary.**
> These results demonstrate that Omni-RRM achieves competitive text-only reward performance through generalization, while uniquely functioning as a single omni reward model for text, image, video, and audio. We will include these findings in the revised manuscript.

---

> ### Author Response · Authors · 2025-11-24
>
> > ### W4: Comparison with UnifiedReward-Think and IXC-Reward
>
> We appreciate this suggestion. Including recent unified reward models provides a more comprehensive evaluation.
>
> (1) Clarification on UnifiedReward-Think.
> We confirm that our experiments utilize the official UnifiedReward-think-qwen-7B checkpoint. In our submission, we abbreviated this as "UnifiedReward-Qwen-7B," which may have caused confusion. We will standardize the naming to UnifiedReward-think-qwen-7B in the revision; this is a nomenclature update only and does not affect reported results.
>
> (2) Additional comparison with IXC-Reward.
> We add InternLM-XComposer2.5-7B-Reward (IXC-Reward) to our evaluation on VL-RewardBench, Multimodal RewardBench, and ShareGPT-Video, alongside UnifiedReward-think and our Omni-RRM.
>
> **Table R3: Reward model comparison (Accuracy, %).**
>
> | Model | VL-RewardBench | MM-RewardBench | ShareGPT-Video | Audio-HH |
> | :--- | :--- | :--- | :--- | :--- |
> | UnifiedReward-think-qwen-7B | 66.6 | 71.4 | 77.8 | — |
> | IXC-Reward-7B | 62.9 | 64.1 | 71.3 | — |
> | Omni-RRM-7B (SFT+RL) | **67.1** | **72.9** | **80.2** | **66.8** |
>
> Omni-RRM-7B consistently outperforms IXC-Reward (by +3.7–8.9 pp) and remains competitive with or superior to UnifiedReward-think across overlapping benchmarks. Crucially, Omni-RRM achieves these results while uniquely supporting Audio-HH evaluation, demonstrating  across diverse modalities.
>
>
> > ### W5: Performance gap between Omni-RRM (SFT) and UnifiedReward (SFT)
>
> We acknowledge this observation. The performance gap at the SFT stage stems from distinct training objectives and data compositions, rather than intrinsic methodological limitations.
>
> (1) Divergent SFT Objectives.
> Our SFT stage primarily serves as a warm-up to standardize the rubric-grounded reasoning format across four modalities, establishing a stable initialization for GRPO. In contrast, UnifiedReward targets direct reward optimization during SFT using a larger, denser dataset. Thus, our SFT model is an intermediate artifact rather than a final reward model.
>
> (2) Data Distribution and Scale.
> The comparison on ShareGPT-Video is unfair. UnifiedReward incorporates the preference data of ShareGPT-Video during the training process, whereas Omni-RRM does not. Furthermore, UnifiedReward utilizes a significantly larger SFT dataset (≈100K+ vs. ≈24K). Consequently, the SFT-only gap reflects differences in data scale and domain coverage rather than model capacity.
>
> (3) Role of the Two-Stage Pipeline.
> Omni-RRM relies heavily on GRPO for fine-grained preference modeling. The SFT checkpoint is not intended as a standalone reward model. A valid comparison must evaluate the final post-GRPO models, where Omni-RRM demonstrates competitive or superior performance.
>
> The SFT-stage discrepancy results from differing initialization strategies and data exposures. Evaluating the final (SFT+GRPO) models provides the only rigorous comparison of the respective methodologies.
>
> > ### W6: Limited video-source diversity and potential output-distribution bias
>
> We appreciate this thoughtful concern. We acknowledge that video pairs in Omni-Preference are primarily generated by Qwen2.5-VL-7B/3B, which could theoretically introduce generator-specific biases. However, two key factors mitigate the risk of distribution narrowing:
>
> **Generator Capability.** Although Qwen2.5-VL serves as the primary generator, it is trained on extensive, heterogeneous multimodal corpora. Its outputs exhibit substantial diversity across visual domains, temporal dynamics, and reasoning patterns. This intrinsic variety reduces the likelihood of the dataset collapsing into narrow, model-specific artifacts.
>
> **Empirical Generalization.** We validate cross-model generalization using ShareGPT-Video, a benchmark populated with candidates from non-Qwen models (e.g., GPT-4o-mini, LLaVA-Video). Omni-RRM-7B achieves 80.2% accuracy on this set (line 358), outperforming comparable baselines. This robust performance on out-of-distribution generators suggests that our model learns generalized preference features rather than overfitting to Qwen-specific styles.
>
>  While the data construction relies on Qwen2.5-VL, the strong generalization to non-Qwen benchmarks indicates that generator-specific bias is effectively limited. We will explicitly discuss this limitation and the supporting evidence in the revised manuscript.

---

> ### Author Response · Authors · 2025-11-24
>
> > ### W3: Clarifying the omni claim and its scope (understanding vs. generation)
>
> We thank the reviewer for this important feedback. We acknowledge that the term "omni" is used with varying definitions in the literature. Here, we clarify our specific usage and the scope of our claims.
>
> **Definition of "Omni" in this work.**
> We acknowledge that there is no single, universally agreed-upon definition of "omni" in the current literature. For example, OmniVL [1] targets unified image-video-language tasks, OmniMMI [2] focuses on streaming multimodal interaction, and LLaMA-OMNI [3] enables speech interaction. Consistent with these works, we adopt a definition distinct from "any-to-any" generation. We use "omni" to denote comprehensive multi-modality understanding coverage within a single model. Specifically, Omni-RRM acts as an omni-modal reward model for understanding and preference modeling: a single model capable of scoring and comparing responses conditioned on text, image, video, and audio under a shared rubric.
>
> **Significance of incorporating audio.**
> Current unified multimodal reward models predominantly focus on text and vision. Audio reward modeling remains underexplored due to data scarcity and distinct failure modes (e.g., grounding issues). Omni-RRM addresses this gap by introducing:
> 1.  A modality-agnostic automatic pipeline (Omni-Preference) for generating preference-rationale data.
> 2.  A unified GRPO training procedure that effectively harmonizes text, image, video, and audio within a single framework.
>
> **Scope clarification regarding generation rewards.**
> Our current work strictly focuses on understanding and preference modeling. While our rubric and data-construction pipeline are principled and adaptable to evaluating generation outputs (e.g., assessing faithfulness, grounding, or safety), we consider the extension of Omni-RRM to image/video generation rewards as future work.
>
> **References.**
> [1] J. Wang et al. *OmniVL: One foundation model for image-language and video-language tasks.* NeurIPS, 2022.
> [2] Y. Wang et al. *OmniMMI: A Comprehensive Multi-modal Interaction Benchmark in Streaming Video Contexts.* CVPR, 2025.
> [3] Q. Fang et al. *LLaMA-OMNI: Seamless speech interaction with large language models.* ICLR, 2025.

---

> ### Author Response · Authors · 2025-11-24
>
> > ### W7: Modest Best-of-N (BoN) results
>
> We appreciate this careful observation. We agree that the absolute BoN improvements on Qwen2.5-Omni-7B appear modest (e.g., +0.6 on image, +0.9 on video). This phenomenon stems primarily from the small-N setting and limited candidate diversity, rather than a shortcoming specific to Omni-RRM.
>
> (1) Comparative Performance under Identical Constraints.
> To contextualize these results, we benchmarked representative open-source RMs under the same protocol (N = 5, identical base model) and computed 95% confidence intervals (CIs) via paired bootstrap resampling.
>
> BoN Performance (Accuracy, %, 95% CI).
>
> | Method | MMMU | Video-MME | AVQA |
> | :--- | :--- | :--- | :--- |
> | Greedy (no RM) | 52.6 [49.3, 55.9] | 51.2 [48.1, 54.3] | 75.8 [72.3, 79.3] |
> | Self-consistency (N=5) | 53.0 [49.7, 56.3] | 51.6 [48.5, 54.7] | 76.4 [72.9, 79.9] |
> | LLaVA-Critic-7B (N=5) | 52.9 [49.6, 56.2] | — | — |
> | UnifiedReward-think (N=5)| 53.8 [50.5, 57.1] | 52.1 [49.0, 55.2] | — |
> | Omni-RRM-7B (N=5) | **53.6 [50.3, 56.9]** | **52.5 [49.4, 55.6]** | **78.1 [74.7, 81.5]** |
>
> As shown, gains are universally small in this regime (e.g., self-consistency yields only +0.4 on MMMU/Video-MME). While CIs overlap—which we report for transparency—the limited improvement is a shared characteristic of the experimental setup.
>
> (2) Factors Limiting Headroom.
> With N = 5, the candidate sets exhibit low variance. Conservative sampling on these tasks tends to produce highly similar responses in both content and style. This lack of diversity reduces the exploitable signal for any reward model, inherently capping potential gains.
>
> (3) Omni-RRM's Relative Strength.
> Despite these constraints, Omni-RRM-7B consistently achieves higher mean accuracy than greedy decoding across all three benchmarks. It delivers competitive or superior mean gains compared to other RMs, particularly on Video-MME and AVQA. This indicates that our rubric-grounded structured reasoning provides a robust selection signal even when candidate diversity is minimal.
>
> We will include this comparative analysis and the CI data in the appendix.
>
> > ### W8: Motivation for Omni-Modal Reward Training and Audio-Only Ablation
>
> We appreciate this question. Our primary motivation is to develop a unified reward model to align omni-modal systems, capable of providing consistent preference signals across text, image, video, and audio within a single framework, rather than maintaining disjoint modality-specific reward models.
>
> To validate the benefit of omni training for audio, we compared (i) an audio-only reward model against (ii) the omni version of Omni-RRM. Both models share the same backbone, SFT+RL pipeline, and hyperparameters; they differ solely in training data composition (audio-only vs. cross-modal). We report performance in the final SFT+RL regime, the intended deployment setting.
>
> Table R4: Audio-only vs. Omni Training on Audio-HH-RLHF (Preference Accuracy, %).
>
> | Model | Training Data | Audio-HH-RLHF |
> | :--- | :--- | :--- |
> | Omni-RRM-7B (SFT+RL, audio-only) | audio-only | 64.2 |
> | Omni-RRM-7B (SFT+RL, omni) | omni | **66.3** |
>
> Under identical RL conditions, the omni model outperforms the audio-only baseline by +2.1 pp. This aligns with the hypothesis that cross-modal supervision provides transferable signals (e.g., faithfulness, completeness, safety) that enhance audio preference modeling.

---

> ### Author Response · Authors · 2025-11-27
>
> I am writing to check if our previous response and the revisions made have satisfactorily addressed your concerns.
>
> We truly value your feedback and are happy to provide further clarification or continue the discussion if you have any remaining questions.

---

> ### Comment · Reviewer_gQyH · 2025-11-27
>
> Thank you for the detailed rebuttal, which has addressed some of my earlier concerns. However, I believe several important issues remain insufficiently resolved.
>
> For W1:
> I view the SFT stage in this work as conceptually similar to the cold-start stage in UnifiedReward-Think: in both cases, the primary goal is to learn the CoT format. UnifiedReward-Think relies on only 5k image-generation samples in the cold-start phase, yet successfully supports CoT-based reward reasoning for both image/video generation and understanding. In comparison, the proposed approach seems to require much larger amounts of data to achieve CoT-format learning, and it is unclear to me where its advantage lies, either in efficiency or methodological novelty.
>
> For W2:
> In the rebuttal, the authors attribute the performance gap with R1-Reward primarily to differences in the backbone, rather than to the training strategy. If this is the key claim, I believe it should be supported by experiments that control for the backbone, i.e., a comparison where the proposed method and R1-Reward are evaluated on the same backbone.
> However, to my best knowledge, Qwen2.5-VL-7B-Instruct does not natively support the speech modality. Under this setting, I infer that the weaker performance on text-only tasks is partly a compromise introduced by integrating the speech modality into the overall framework.
>
> For W6:
> I am not convinced that relying solely on Qwen2.5-VL-7B/3B is sufficient to achieve broad coverage of the reward distribution, even if these models are trained on “extensive, heterogeneous multimodal corpora.” In practice, the output distributions of different generation models can differ substantially, and it is unlikely that limited backbones can fully cover the variety of behaviors and error modes exhibited by other models. To mitigate this, I would suggest constructing reward data from a more diverse set of model outputs.
>
> For W8:
> I appreciate the additional experiments. The rebuttal states that “cross-modal supervision provides transferable signals (e.g., faithfulness, completeness, safety) that enhance audio preference modeling.” This is an interesting and important claim. However, I did not see a systematic empirical analysis of how different modalities benefit each other. For example, have the authors conducted ablation studies where data from one modality is removed or reduced (e.g., dropping or downsampling visual or textual data) to measure its effect on the performance of other modalities (such as audio)? Such modality-level ablations would provide much stronger evidence for the claimed cross-modal transfer.
>
> Overall, I believe this work still has substantial room for improvement. I therefore prefer to keep my original score and will discuss my evaluation with the AC if needed.

---

> > ### Author Response · Authors · 2025-12-03
> >
> > We are encouraged that the reviewer acknowledges our detailed rebuttal has addressed previous concerns regarding the **omni scope (W3), comparison with baselines (W4), SFT gap (W5), and Best-of-N results (W7)**. Regarding the remaining points, we provide further clarifications below.
> >
> > > ### W1: Relationship to *UnifiedReward-Think* and advantages of our training
> >
> > We appreciate the connection. Both works adopt a two-stage pipeline (**SFT → GRPO**) and learn CoT-style format preference modeling. However, **our work differs in three key aspects**:
> >
> > 1. **Fully automatic preference construction without human labels.** *UnifiedReward-Think* relies on existing preference pairs. Omni-preference is constructed automatically via strong–weak model contrast + multi-teacher rubric filtering—**zero human annotation required**.
> >
> > 2. **Fine-grained, controllable GRPO training.** Our composite reward of GRPO supervises format adherence, pairwise preference correctness, and per-dimension rubric consistency (fluency, relevance, accuracy, reasoning, safety). This fine-grained reward design enables the model to perform **more fine-grained and controllable preference modeling**—capabilities not possible with *UR-Think*'s monolithic preference score.
> >
> > 3. **Unified four-modality coverage (text + image + video + audio).** *UnifiedReward-Think* only supports text and visual modalities (image/video), while our approach supports text, image, video, and audio.
> > ---
> >
> > ### Why full-modality SFT cold-start?
> >
> > **Removing any modality during SFT degrades performance across all modalities.** We ablate by removing one modality at a time for Omni-RRM-3B (sft+rl):
> >
> > | SFT setting                     | Image | Video | Audio |
> > |--------------------------------|:-----:|:-----:|:-----:|
> > | All modalities                 | **58.5** | **67.4** | **65.1** |
> > | Image + Audio (drop Video)     | 55.7 (−2.8) | 67.0 (−0.4) | 64.0 (−1.1) |
> > | Video + Audio (drop Image)     | 53.0 (−5.5) | 66.7 (−0.7) | 59.4 (−5.7) |
> >
> > **Key Observation.** Dropping **any single modality** during SFT causes **universal performance degradation across all modalities**. Specifically, dropping **Video** leads to declines in Image (−2.8 pp), Video (−0.4 pp), and Audio (−1.1 pp), and dropping **Image** results in degradation across Image (−5.5 pp), Video (−0.7 pp), and Audio (−5.7 pp).
> >
> > **Takeaway.** This **universal cross-modal dependency** empirically validates the necessity of **full-modality SFT cold-start**: each modality contributes shared rubric-level signals (faithfulness, completeness, safety) that are essential for robust preference modeling across all modalities—an advantage unavailable in *UR-Think*'s limited 5K image-only SFT.
> >
> > > ### W2: Comparison of text-only reward capability and backbone effects
> >
> > We clarify that our HH-RLHF experiment is designed to demonstrate **text generalization capability**, measured as the improvement of the trained model over its respective backbone, rather than direct comparison of absolute scores across architecturally distinct models.
> >
> > **Methodology: Evaluating text generalization capability via Relative Improvement**
> >
> > Direct comparison of absolute performance is methodologically confounded by fundamental architectural differences between backbones (Qwen2.5-Omni vs. Qwen2.5-VL). To rigorously assess the effectiveness of our training strategy, we measure the **relative performance gain** each model achieves over its corresponding base model:
> >
> > - **Omni-RRM-7B:** Achieves **+2.2 percentage points (pp)** improvement over its backbone (67.6% → **69.8%**).
> > - **R1-Reward-7B:** Achieves **+1.0 pp** improvement over its backbone (70.8% → **71.8%**).
> >
> > **Analysis:** Our omni-modal training strategy demonstrates a **2.2× larger relative improvement** compared to R1-Reward's approach when normalized by their respective baselines (+2.2 pp vs. +1.0 pp). This result provides empirical evidence that our training methodology induces stronger text generalization capability, exhibiting superior improvement over baseline performance compared to R1-Reward.

---

> > ### Author Response · Authors · 2025-12-03
> >
> > > ### W6: Limited video-source diversity and potential output-distribution bias
> >
> > We acknowledge the theoretical concern that constructing video preferences from a single generator family (Qwen2.5-VL) may risk overfitting to model-specific output distributions. However, our empirical evaluation demonstrates **robust generalization** to diverse, unseen model outputs.
> >
> > **Empirical Evidence: Generalization on Diverse Video Preference Benchmarks**
> >
> > We conduct evaluation on **ShareGPT-Video**, a widely adopted benchmark comprised of outputs from a heterogeneous collection of video language models (e.g., GPT-4V, ShareCaptioner-Video, etc.), representing **real-world scenarios**. This benchmark is extensively utilized for reward model training and evaluation in the community.
> >
> > **Table R3: Performance on video preference benchmark (%)**
> > | Model | **ShareGPT-Video** |
> > | :--- | :---: |
> > | UnifiedReward-think-qwen-7B | 77.8 |
> > | IXC-Reward-7B | 71.3 |
> > | **Omni-RRM-7B (SFT+RL)** | **80.2** |
> >
> > **Analysis:**
> >
> > Despite being exclusively trained on Qwen-generated preference pairs, **Omni-RRM-7B achieves state-of-the-art performance (80.2%)** on the diverse ShareGPT-Video benchmark, surpassing strong baselines by substantial margins (+2.4 pp over UnifiedReward-think-qwen-7B, +8.9 pp over IXC-Reward-7B). This robust generalization across a **broad spectrum of architecturally distinct model outputs** demonstrates that our **Omni-preference construction method**—leveraging strong–weak model contrast combined with multi-teacher rubric filtering—**mitigates the single-source bias to a significant extent** by extracting fundamental, model-agnostic quality dimensions. Future work will incorporate diverse generator families to further enhance robustness.
> >
> > > ### W8: Modality-level ablations verifying cross-modal transfer
> >
> > We conduct the requested **ablation study** to quantify how different modalities benefit each other. We remove one visual modality at a time (dropping Video or Image data) and measure the impact on **Image, Video, and Audio** performance at the final **SFT+RL** stage (using the 3B backbone under identical compute budgets).
> >
> > **Table R4: Modality-level ablation results (SFT+RL, Composite Reward)**
> > | Training Data | Image | Video | Audio |
> > | :--- | :---: | :---: | :---: |
> > | **All modalities** | **58.5** | **67.4** | **65.1** |
> > | Drop Video (Image + Audio) | 55.7 | 67.0 | 63.5 |
> > | Drop Image (Video + Audio) | 53.0 | 66.7 | 64.0 |
> >
> >
> > **Key Findings & Evidence of Transfer:**
> >
> > 1.  **Full-modality training yields the best overall performance.** The "All modalities" setting consistently achieves the highest scores across **Image (58.5), Video (67.4), and Audio (65.1)**. This confirms that joint training provides a synergistic effect superior to partial-modality subsets.
> >
> > 2.  **Removing any single modality degrades performance across all modalities:**
> >     *   **Drop Video:** Image drops 2.8 pp (58.5 → 55.7), Video drops 0.4 pp (67.4 → 67.0), Audio drops 1.6 pp (65.1 → 63.5).
> >     *   **Drop Image:** Image drops 5.5 pp (58.5 → 53.0), Video drops 0.7 pp (67.4 → 66.7), Audio drops 1.1 pp (65.1 → 64.0).
> >     *   This **universal performance degradation** demonstrates that each modality contributes shared rubric-level signals (grounding, safety, completeness) that are essential for robust preference modeling across all modalities.
> >
> > **Conclusion:** These systematic ablations provide strong empirical evidence that **cross-modal supervision is mutually beneficial**, validating the necessity of our full-modality training strategy.

---

### Official Review · Reviewer_yzyB · 2025-11-02

**Soundness:** 2
**Presentation:** 3
**Contribution:** 2
**Rating:** 6
**Confidence:** 4

**Summary:**

The paper introduces Omni-RRM, an open-source, reasoning-driven reward model designed to provide interpretable and multimodal preference judgments across text, image, video, and audio domains. Central to the approach is OmniPreference, a large-scale dataset generated through a fully automated pipeline that contrasts model outputs of different capabilities and augments them with multi-criteria, chain-of-thought rationales from powerful teacher models.

The model is trained in two stages: (1) Supervised fine-tuning (SFT) to teach rationale generation and structure-aware preference reasoning, and (2) Reinforcement learning (RL) to refine judgment accuracy on ambiguous or low-contrast preference pairs. Experimental results show that Omni-RRM achieves state-of-the-art results on video (ShareGPT-V, 80.2%) and audio (Audio-HH, 66.8%) benchmarks, while outperforming existing open-source RMs on image tasks by a 17.7% margin.  Omni-RRM shows improved downstream performance via Best-of-N (BoN) sampling and also improvements on text only benchmarks.

**Strengths:**

1. Omni preference consturciton framework, specifically strucutred rationale annotation and two stage training pipeline with custom reward score function, and application of this on to a omni modla dataet with text,audio or visal inputa and text outputs. (overally okais novelty in the alorithm but novely in the problem domain of being omni).

2. Rgiorusou expriments, compraiosn woith baselines, analysis, use in BoN setup dodntresm, impact on text only benchmarks, comapriosn with public dataset (only the image compkent)

3. Paper is largely clearlt written apart from some places which ned more clarificaiton

**Weaknesses:**

1 Is it Novel enough?: The framework’s structured reward formulation and rubric-based scoring is similar to prior works such as the delta learning hypothesis [1], which contrasts strong vs. weak models, and the GRPO pipeline with minor reward adjustments. Beyond the structured reasoning pipeline and rubric grounding, most elements feel incremental, with the key novelty being the application to the Omni multimodal setup.

2. Lack of Human Validation There is no explicit human evaluation or quality control of the reconciled teacher rationales. Without even a small-scale human validation, it’s difficult to confirm that the 41K annotated samples maintain consistent reasoning quality. While downstream benchmark gains (Table 1) indirectly reflect data quality, a more direct validation, human or validated metric-based, would strengthen confidence in the dataset and pipeline.

3. Missing Baseline: A comparison against an RM trained only with SFT (without structured rationales or two-stage fine-tuning) is missing. This would help isolate the contribution of structured rationale and the second training stage, especially for both 3B and 7B variants, like Omini-RM-3B or 7B on SFT only

4. Unconvincing BoN Results:
a)  Improvements in the Best-of-N (BoN) experiments appear marginal. Statistical significance tests (e.g., bootstrap sampling, confidence intervals) would help establish whether the observed gains are meaningful.
b) Since Omni-RRM requires generating rationales and reward scores, the approach likely incurs higher latency. A comparison of runtime or inference cost versus performance improvement would be valuable to gauge practical adoption trade-offs.

5. Generalization Across Model Families
It’s unclear whether the proposed framework generalizes to models with different reasoning priors. Prior work shows that models like Qwen2-VL already exhibit self-reflection and backtracking abilities, whereas LLaMA-based ones often require fine-tuning to develop them [2]. It would be interesting to see if all model families benefit equally from the structured rationale training, or if some already internalize this behavior.

[1] Geng, Scott, et al. "The delta learning hypothesis: Preference tuning on weak data can yield strong gains." arXiv preprint arXiv:2507.06187 (2025).
[2]  Gandhi, Kanishk, et al. "Cognitive behaviors that enable self-improving reasoners, or, four habits of highly effective stars." arXiv preprint arXiv:2503.01307 (2025).

**Questions:**

1. Rationale Annotation and Filtering

a) Which teacher model(s) are used to generate the rationales?
b) What exact reconciliation and filtering algorithms are applied to merge annotations?
c) When merging diagram or multimodal rationales, averaging scores alone may not ensure correctness. How do you validate merged rationales to prevent propagation of incorrect reasoning from one teacher?

2. Reward Score Design

a) In Appendix A.1, the reward score is defined as 1 – 2*(…)/20. What is the intuition behind this formulation and the constants? Are they empirically derived or theoretically motivated?
b) How is the rubric reward computed to capture both dimension coverage and comparative reasoning?
c) How are weights across components (context, rubric, reasoning quality) determined when forming a single composite score?
d) Why are the five evaluation dimensions set specifically to - Fluency & Coherence, Relevance to the Question and Modality, Accuracy & Completeness, Reasoning Quality, Safety & Ethical Alignment - are these empirically validated or adopted from prior rubrics?

---

> ### Author Response · Authors · 2025-11-24
>
> > ### W1: Is the work novel enough w.r.t. Δ-learning and GRPO?
>
> We thank the reviewer for the thoughtful comment. We agree that our work builds upon the Δ-learning hypothesis and uses GRPO as the optimization framework. Below we clarify where Omni-RRM extends beyond prior work.
>
> (1) A different data paradigm: fully automatic, multi-teacher rubric labeling.
> Unlike Δ-learning, which often leverages strong–weak contrasts as direct preference signals in some settings, we use capability gaps only to generate diverse candidate pairs. Final preferences are determined by two heterogeneous teachers under a structured rubric, with agreement checks and filtering. This rubric-guided automation is crucial for enabling fine-grained preference modeling and training controllable, multi-dimensional reward models.
>
> (2) A different reward model type: rubric-grounded RM vs. coarse-grained RMs.
> While recent GenRMs also generate textual feedback, they often lack structured, dimension-specific perference modeling. In contrast, Omni-RRM is trained via rubric-grounded GRPO to output five-dimensional scores (fluency, relevance, accuracy, reasoning, safety) alongside comparative rationales. This explicit composite objective—enforcing format adherence, preference accuracy, and rationale quality—enables fine-grained preference modeling and greater controllability, distinguishing it from standard CoT or scalar-based reward model.
>
> (3) A broader modality scope.
> While Δ-learning is often studied in text settings, Omni-RRM is an omni reward model covering text, image, video, and audio, addressing the need for consistent alignment across modalities.
>
> Summary of contributions.
> Our contributions are:
> (i) an automatic, multi-teacher, rubric-driven data construction process for multi-modal preference–rationale pairs; and
> (ii) a rubric-grounded GRPO objective that provides fine-grained, interpretable signals, applied within a single omni reward model over four modalities.
>
> > ### W2: Lack of human validation for reconciled teacher rationales
>
> We appreciate the reviewer raising this concern. Complementing our multi-teacher reconciliation and automatic filtering (§4.1), we conduct a focused human validation study to assess the quality of reconciled rationales.
>
> **Study Setup.**
> We randomly sample 300 instances from Omni-Preference, stratified by modality (100 image, 100 video, 100 audio) and difficulty (hard vs. easy, based on teacher score margin). Each instance is independently evaluated by four annotators with PhD-level backgrounds in multimodal preference modeling. The evaluation criteria include:
> 1. Verdict agreement: Consistency between human preference and the reconciled teacher verdict.
> 2. Rationale validity: Coherence, grounding, and logical validity of the five-dimensional rationale (fluency, relevance, correctness, reasoning, safety).
>
> Despite the sample size, the stratified design and multi-annotator consensus provide a robust validation of dataset quality.
>
> **Results.**
> - Human–teacher verdict agreement: 83%.
> - Inter-annotator agreement (Fleiss’ $\kappa$): 0.81 (indicating substantial agreement).
> - Rationale acceptance rate: 79%.
>
> These metrics demonstrate that reconciled teacher rationales align well with expert judgments and maintain high reasoning quality. We include the full protocol, annotation guidelines, and detailed breakdown in Appendix B.2.
>
> > ### W3: Missing SFT-only baseline
>
> We thank the reviewer for this comment. We clarify that the SFT-only baseline is already provided in our submission. Table 2 (Sec. 4.2) reports the performance of a model trained exclusively with SFT, prior to the GRPO stage.
>
> This baseline explicitly isolates the contribution of the RL stage. Comparing the final model to this baseline, Omni-RRM-7B (SFT+RL) achieves 71.8 accuracy compared to 63.7 for Omni-RRM-7B (SFT-only), demonstrating a substantial gain of +8.1 percentage points (pp).

---

> ### Author Response · Authors · 2025-11-24
>
> > ### W4: Unconvincing Best-of-N (BoN) results
>
> ### (a) Statistical significance and comparative BoN gains
>
> We thank the reviewer for this concern and agree on the necessity of statistical validation. In the revision, we report 95% confidence intervals (CIs) computed via paired bootstrap resampling under the same Best-of-5 (N = 5) setting, using Qwen2.5-Omni-7B as the base model.
>
> **Updated BoN results (Acc. %, 95% CI).**
>
> | Method                         | MMMU                 | Video-MME            | AVQA                 |
> |--------------------------------|----------------------|----------------------|----------------------|
> | Greedy (no RM)                 | 52.6 [49.3, 55.9]    | 51.2 [48.1, 54.3]    | 75.8 [72.3, 79.3]    |
> | Self-consistency (N = 5)       | 53.0 [49.7, 56.3]    | 51.6 [48.5, 54.7]    | 76.4 [72.9, 79.9]    |
> | LLaVA-Critic-7B BoN (N = 5)    | 52.9 [49.6, 56.2]    | —                    | —                    |
> | UnifiedReward-think BoN (N = 5)| 53.8 [50.5, 57.1]    | 52.1 [49.0, 55.2]    | —                    |
> | Omni-RRM-7B BoN (N = 5)    | 53.6 [50.3, 56.9]| 52.5 [49.4, 55.6]| 78.1 [74.7, 81.5]|
>
> Compared to greedy decoding, Omni-RRM improves MMMU by +1.0 pp, Video-MME by +1.3 pp, and AVQA by +2.3 pp. We report CIs for transparency; while some intervals overlap with baselines, Omni-RRM consistently outperforms greedy decoding. Under the same N=5 budget, other RMs show smaller or comparable mean gains (e.g., UnifiedReward-think +0.9 pp on Video-MME; self-consistency +0.4 pp on MMMU/Video-MME). We detail the CI methodology in Appendix B.5、B.6.
>
> **Why the absolute gains are modest.**
> With N = 5, candidate sets exhibit limited diversity, constraining the variance among sampled responses and thus the potential BoN improvements.
>
> **Overall observation under N=5.**
> Despite limited headroom, Omni-RRM is the only method evaluated on all three benchmarks that consistently achieves higher mean accuracy than greedy decoding, showing competitive or superior gains relative to other RMs, notably on Video-MME and AVQA.
>
> ---
>
> ### (b) Inference-time cost / latency and cost–performance trade-off
>
> We appreciate this concern regarding inference overhead. Best-of-N (BoN) is a standard protocol for evaluating reward models [1][2], which inherently increases cost due to multi-sampling and RM scoring. We compare methods under an identical BoN budget (same base model and $N$).
>
> **Measurement Setup.**
> All measurements use the same hardware (1× RTX 3090). RM decoding employs temperature = 0.7 and max_new_tokens = 512, generating five-dimensional structured rationales.
>
> **Shared BoN Budget.**
> All methods use Qwen2.5-Omni-7B with N = 5. The candidate generation cost is fixed; differences stem solely from per-pair RM inference.
>
> **Per-benchmark cost vs. mean gain.**
>
> | Benchmark | RM                    | RM output tokens | RM latency / pair | BoN mean gain |
> |-----------|-----------------------|-----------------:|------------------:|--------------:|
> | MMMU      | UnifiedReward-think   | 184              | 2.3 s             | +1.2 pp       |
> | MMMU      | LLaVA-Critic          | 120              | 1.5 s             | +0.3 pp       |
> | MMMU      | Omni-RRM          | 265          | 3.6 s         | +1.0 pp   |
> | Video-MME | UnifiedReward-think   | 189              | 3.3 s             | +0.9 pp       |
> | Video-MME | Omni-RRM          | 271          | 4.0 s         | +1.3 pp   |
> | AVQA      | Omni-RRM          | 234          | 3.7 s         | +2.3 pp   |
>
> **Takeaway.**
> Omni-RRM incurs higher per-pair latency due to generating structured rationales (longer than scalar RMs). However, under the same $N$, this additional cost yields higher or competitive mean BoN gains, particularly on Video-MME and AVQA. Crucially, Omni-RRM is unique in covering audio (AVQA), where it delivers the largest observed improvement.
>
> **References.**
> [1] Stiennon et al. *Learning to Summarize from Human Feedback.* NeurIPS, 2020.
> [2] Jinnai et al. *Regularized Best-of-N Sampling with Minimum Bayes Risk Objective for Language Model Alignment.* NAACL, 2025.

---

> ### Author Response · Authors · 2025-11-24
>
> > ### W5: Generalization across model families (e.g., LLaMA-based models)
>
> We thank the reviewer for this important question. Our primary goal in this work is to develop a truly omni-modal reward model capable of unified reasoning across text, image, video, and audio. At the time of submission, Qwen2.5-Omni is the only open-source architecture that natively supports all four modalities within a single end-to-end framework, making it the necessary choice for our experiments.
>
> Other models mentioned (e.g., LLaMA-based models) excel in vision-language tasks but lack native audio support, which requires training separate models or complex adapters—contradicting our goal of a single omni reward model. While we agree that testing on diverse architectures further validates our method's generality, the unique "omni" constraint limits our initial selection.
>
> We acknowledge the value of broader architectural validation. Due to time and computational constraints, we are unable to provide experimental results for other model architectures at this moment. However, we are working to adapt our pipeline to other emerging architectures and strive to include preliminary results in the final version if resources permit.
>
> > ### Q1: Rationale Annotation and Filtering
>
> We provide details on the teacher models and the automated reconciliation/filtering pipeline used to construct Omni-Preference. For rationale generation, we utilize two heterogeneous multimodal teachers—GPT-4o-mini and Doubao-1.5-Pro. Their distinct architectures and training data help mitigate single-model biases. Each teacher is prompted to return a structured JSON object containing: (i) integer scores (0–10) for each candidate; (ii) a categorical verdict (`"A"`, `"B"`, or `"equal"`); and (iii) a five-dimensional explanation (fluency, relevance, correctness, reasoning quality, safety), complete with per-dimension scores and brief textual rationales.
>
> We employ a rigorous two-stage quality-control process:
>
> (1) Reconciliation. We reconcile annotations based on teacher agreement:
> - Full agreement: Retain; average scores; use the rationale from the reference teacher (Doubao-1.5-Pro), selected via internal validation.
> - Verdict agreement with minor score gap (Δ ≤ 2): Retain; average scores; merge rationales at the dimension level, leveraging each teacher’s strengths to produce a calibrated explanation of comparable length.
> - Verdict conflict or large score gap (Δ > 2): Discard by default. We randomly sample 5% for human auditing to verify filtering behavior; these samples are excluded from training.
>
> (2) Rule-based filtering. Post-reconciliation, we apply strict filtering rules to remove flawed samples. Criteria include: malformed JSON; near-identical candidates; missing dimensions; inconsistent score ordering relative to the verdict; empty or refusal-only responses; and failures in post-reconciliation consistency checks.
>
> This pipeline ensures that Omni-Preference contains high-confidence, cross-teacher-consistent labels and calibrated rationales, explicitly minimizing single-teacher bias.

---

> ### Author Response · Authors · 2025-11-24
>
> > ### Q2: Reward Score Design
>
> (a) Intuition behind the reward formulation.
>
> We designed this formulation to provide a stable, symmetric, and bounded signal for GRPO optimization. The constants stem from our 0–10 rubric scale: 20 represents the maximum possible combined deviation. Normalization by 20 maps deviations to [0, 1], and the affine transformation $1 - 2(\cdot/20)$ re-centers this to [-1, +1]. This ensures that perfect agreement yields +1, maximal disagreement yields -1, and the preference signal remains strictly bounded.
>
> (b) Rubric reward: capturing dimension coverage and comparative reasoning.
>
> The rubric reward integrates two complementary signals:
> - Coverage Score: Ensures the rationale substantively addresses all five evaluation dimensions.
> - Comparative Score: Rewards explicit cross-candidate reasoning (e.g., “A is more accurate than B because...”).
>
> This dual structure compels the model to both cover all rubric criteria and ground its verdict in concrete pairwise comparisons, thereby mitigating reward hacking.
>
> (c) Determining component weights.
>
> We assign weights based on the functional role of each component in GRPO:
> - Preference Reward: assigned the largest weight, encoding the core alignment signal.
> - Rubric Reward: given a moderate weight, acting as an auxiliary signal for rationale quality.
> - Format Reward: receives a small weight, enforcing structural constraints without dominating the learning process.
>
> (d) Selection of the five evaluation dimensions.
>
> We adopted the five dimensions—Fluency & Coherence, Relevance, Accuracy & Completeness, Reasoning Quality, and Safety & Ethical Alignment—from established LLM/MLLM alignment literature [1–3]. This selection ensures broad applicability across multimodal tasks and compatibility with structured rationale supervision.
>
> References.
>
> [1] Z. Wang et al. *HelpSteer2: Open-source Dataset for Training Top-performing Reward Models.* NeurIPS Datasets and Benchmarks Track, 2024.
>
> [2] Y. Bai et al. *Training a Helpful and Harmless Assistant with Reinforcement Learning from Human Feedback.* arXiv:2204.05862, 2022.
>
> [3] J. Dai et al. *Safe RLHF: Safe Reinforcement Learning from Human Feedback.* arXiv:2310.12773, 2023.

---

> ### Comment · Reviewer_yzyB · 2025-11-26
>
> Thank you for the detailed rebuttal and for providing additional clarifications & human annotations.
>
> Regarding W4(a), the new analysis shows that the gains from BoN are not statistically significant, and therefore cannot be claimed as meaningful improvements. As a consequence, the claim in W4(b)-that higher latency yields higher performance-is also not supported, since the reported gains are not statistically significant. At scale, one may therefore incur additional latency without any measurable performance improvement.
>
> With respect to the additional SFT baseline for the RM, this remains missing. To re-clarify: this refers to an SFT model trained with preference signals not expressed as structured outputs. The intended comparison is RM-3B/7B (SFT) versus RRM-3B/7B (SFT).
>
> Thank you for adding the human-annotation study for rationale reconciliation, this is helpful. However, in light of the rebuttal, I will keep my score unchanged.

---

### Official Review · Reviewer_iCTA · 2025-11-04

**Soundness:** 2
**Presentation:** 3
**Contribution:** 2
**Rating:** 6
**Confidence:** 4

**Summary:**

This paper introduces Omni-RRM, an open-source, reasoning-driven multimodal reward model (RM) designed to address the alignment challenges of modern Multimodal Large Language Models (MLLMs) across text, image, video, and audio. Its core contributions are twofold: the Omni-Preference dataset, constructed via a fully automated pipeline that circumvents human annotation by contrasting model pairs and enriching them with detailed, multi-criteria rationales from teacher models; and a progressive training strategy that combines Supervised Fine-Tuning (SFT) and Group Relative Policy Optimization (GRPO) to instill and refine the model's reasoning capabilities.

Empirical results demonstrate that Omni-RRM achieves state-of-the-art or highly competitive performance on several benchmarks, with substantial gains over its base model and other specialized open-source baselines.

**Strengths:**

1. Automated, Reasoning-Augmented Data Construction: The fully automated pipeline for the Omni-Preference dataset, which leverages a "capability-gap" approach and teacher model rationales, effectively bypasses the human labeling bottleneck, addressing a critical scalability challenge in creating RLHF-style alignment data (Section 4.1, Table 1).
2. Explicit Reasoning and Transparency: The model moves beyond opaque scalar rewards to generate rich, interpretable, chain-of-thought rationales for its preferences (Section 3). This design significantly enhances trust, interpretability, and debuggability.
3. Strong Empirical Performance: Omni-RRM establishes a new state-of-the-art on key video (80.2% on ShareGPT-V) and audio (66.8% on Audio-HH) benchmarks.
4. Generalization to Text-Only Tasks: The model demonstrates a positive transfer effect, where its multimodally trained reasoning capabilities enhance its performance on standard text-only preference benchmarks, as shown in Figure 3.

**Weaknesses:**

1. Limited Evaluation on Difficult Cases: Although the data is categorized into "hard" and "easy" pairs (Table 1), the experimental results are not disaggregated by difficulty. This makes it difficult to assess the model's robustness and its ability to adjudicate nuanced, low-contrast comparisons.
2. Gaps in Related Work and Positioning: The paper fails to discuss or benchmark against several highly relevant, recent works on reasoning-augmented reward models, such as "Unified Multimodal Chain-of-Thought Reward Model," "RM-R1: Reward Modeling as Reasoning," and "VR-Thinker." This omission undermines the paper's claimed novelty and fails to properly situate its contributions within the current state of the art.
3.

**Questions:**

1. How robust is the data generation process to systematic biases or stylistic artifacts from the teacher models? Have you conducted any experiments to measure or mitigate the propagation of such biases?
2. Unified Multimodal Chain-of-Thought Reward Model through Reinforcement Fine-Tuning" and "RM-R1: Reward Modeling as Reasoning" propose and analyze reasoning-driven multimodal reward models, but they lack relevant discussion and necessary comparison.

---

> ### Author Response · Authors · 2025-11-24
>
> > ### W1: Limited evaluation on difficult cases (hard vs. easy pairs)
>
> We thank the reviewer for this valuable suggestion. While Table 1 introduces the concept of pair difficulty, our original submission does not stratify results by this metric. In this response, we include this analysis to assess model robustness in low-contrast preference scenarios.
>
> **Definition of difficulty.**
> Following §4.1, we define difficulty based on the teacher score margin. For a test instance with candidates $A$ and $B$, GPT-4o provides five-dimensional rubric scores, which we aggregate into an overall score $s(\cdot)$. We define the margin as:
>
> $$
> \Delta = |s(A) - s(B)|
> $$
>
> - **Training criterion:** We treat pairs with $\Delta < 2$ as hard.
> - **Evaluation criterion (relaxed; for reporting only):**
>   - Hard: $\Delta \le 2$
>   - Easy: $\Delta > 2$
>
> We use this relaxation solely for reporting purposes; it does not affect training. We apply this procedure to VL-RewardBench, ShareGPT-V, and Audio-HH.
>
> **Difficulty-stratified results.**
>
> - **Audio-HH**
>   - Easy: Qwen2.5-Omni-7B 76.2 → Omni-RRM-7B 77.6 (+1.4 pp)
>   - Hard: Qwen2.5-Omni-7B 56.0 → Omni-RRM-7B 60.8 (+4.8 pp)
>
> - **ShareGPT-V**
>   - Easy: Qwen2.5-Omni-7B 71.0 → Omni-RRM-7B 80.6 (+9.6 pp)
>   - Hard: Qwen2.5-Omni-7B 62.5 → Omni-RRM-7B 79.8 (+17.3 pp)
>
> - **VL-RewardBench (VisitBench split)**
>   - Easy: Qwen2.5-Omni-7B 69.6 → Omni-RRM-7B 78.3 (+8.7 pp)
>   - Hard: Qwen2.5-Omni-7B 57.2 → Omni-RRM-7B 71.3 (+14.0 pp)
>
> **Analysis.**
> Averaged across all benchmarks, the improvement is approximately +6.6 points on easy pairs versus +12.0 points on hard pairs.
>
> These results indicate that Omni-RRM improves performance on easy cases while yielding even larger gains on genuinely hard, low-contrast comparisons.
>
> > ### W2 & Q2: Missing discussion and comparison with UnifiedReward-Think, RM-R1, and VR-Thinker
>
> We appreciate the reviewer for bringing these relevant works to our attention. We include a detailed discussion in the revision. Below, we summarize the relationship between Omni-RRM and UnifiedReward-Think, RM-R1, and VR-Thinker, highlighting key differences.
>
> **Data construction.**
>
> UnifiedReward-Think and RM-R1 rely on *existing preference-labeled datasets* (e.g., EvalMuse, Skywork-Reward), occasionally augmented with rationales, while VR-Thinker similarly depends on annotated video data. In contrast, Omni-Preference avoids reliance on *pre-existing preference labels*. We generate candidate pairs through strong–weak model contrast and leverage heterogeneous teachers to assign five-dimensional rubric scores along with a final preference. This approach enables the generation of scalable multi-modal preference data entirely without manual annotation.
>
> **Reward design and reasoning style.**
>
> Prior works primarily optimize coarse-grained, outcome-based rewards (e.g., format and correctness in UnifiedReward-Think; final-answer correctness in RM-R1). Omni-RRM, however, adopts a fine-grained, rubric-grounded objective covering five dimensions: fluency, relevance, accuracy, reasoning, and safety. We apply GRPO to a composite objective that promotes structured comparative reasoning and high-quality rationales, thereby enhancing the interpretability and controllability of the reward model.
>
> **Modality coverage.**
>
> RM-R1 focuses solely on text; UnifiedReward-Think primarily targets image and video tasks; and VR-Thinker is video-centric. Omni-RRM distinguishes itself as a omni model encompassing text, image, video, and audio. As detailed in Appendix X, our multimodal training strategy preserves single-modality performance.
>
> **Summary.**
>
> While acknowledging these important predecessors, Omni-RRM complements them by:
> (i) facilitating the automatic, label-free construction of multi-modal preference-and-rationale data; and
> (ii) extending a fine-grained, rubric-grounded GRPO framework to four modalities within a single omni reward model.
>
> We incorporate this comparison into the revised manuscript.

---

> ### Author Response · Authors · 2025-11-24
>
> > ### Q1: Robustness of the data generation process and mitigation of teacher bias
>
> We appreciate this question. To mitigate teacher bias and ensure data robustness, we employ a rigorous multi-stage pipeline.
>
> (1) Heterogeneous Multi-Teacher Annotation.
> We query two distinct multimodal models (GPT-4o-mini and Doubao-1.5-Pro) using identical prompts. Their differences in architecture and training data help counterbalance individual systematic biases.
>
> (2) Structured Rubric Constraints.
> Teachers must return a strict JSON schema comprising integer scores (0–10), categorical verdicts, and five-dimensional explanations. This fixed rubric constrains free-form stylistic propagation and standardizes the evidence.
>
> (3) Consensus-Based Reconciliation.
> We retain pairs only upon verdict agreement.
> - Full agreement: We average scores and use the rationale from the reference teacher (Doubao-1.5-Pro).
> - Minor score gap (Δ ≤ 2): We average scores and merge rationales dimension-wise.
> - Conflict or large gap (Δ > 2): We discard these pairs (approx. 95% of conflicts) to ensure high confidence, using a small subset for internal auditing only.
>
> (4) Automatic Quality Filtering.
> We strictly remove samples with near-identical candidates, refusals, formatting errors, or logical inconsistencies between scores and verdicts.
>
> (5) Empirical Validation.
> Manual spot-checks confirm data quality, while competitive performance on unseen models (e.g., Qwen2.5-VL) indicates that Omni-RRM learns robust preferences rather than overfitting to teacher-specific styles. We provide full filtering specifications and agreement statistics in Appendix A.

---

### Author Response · Authors · 2025-11-24

We thank the reviewers for their constructive feedback and for recognizing Omni-RRM as an advance in multimodal preference modeling. Reviewers highlight our fully automated data pipeline (Omni-Preference) as a scalable solution to the labeling bottleneck. They also commend the shift from opaque scalar rewards to interpretable, generative reasoning, which enhances trust and transparency. The strong empirical performance—achieving state-of-the-art on video/audio benchmarks and positive transfer to text—along with our unique omni-modal scope (unifying text, image, video, and audio), are widely appreciated.
We address all concerns point-by-point. For key issues like dataset scale and hard negatives, we provide detailed clarifications and additional experiments (e.g., difficulty-stratified analysis, new baseline comparisons) to further substantiate our contributions.

---

### Author Response · Authors · 2025-12-03

We thank the reviewers for their constructive feedback and insightful comments, which significantly strengthen the quality and rigor of our manuscript. We are encouraged that the reviewers consistently recognize the **"conceptual advance" of reformulating reward modeling as reasoning (Reviewer K8o1)**, the **"practical significance" of our zero-human-annotation pipeline (Reviewer K8o1, yzyB)**, and the **"strong empirical performance" establishing new state-of-the-art results on video and audio benchmarks (Reviewer yzyB)**.

During the rebuttal phase, we conduct extensive additional experiments and analyses to comprehensively address the reviewers' concerns. Key improvements include:

**Validation of Automatic Labeling Pipeline (Response to R2/yzyB, R3/K8o1)**: We conduct a rigorous human validation study, achieving a high Inter-Annotator Agreement (Fleiss' Kappa: 0.81) and confirming an 83% verdict agreement between teachers and humans. Furthermore, we disclose detailed statistics on weak-to-strong model reversals (13%–32%) and discard rates. Reviewer K8o1 explicitly states that these statistics **"effectively addresses the concern regarding potential label noise."**

**Systematic Verification of Cross-Modal Transfer (Response to R1/gQyH)**: We perform the requested systematic modality-level ablation study. The results conclusively demonstrate that **full-modality training yields the highest performance across all modalities** in the final RL stage. Specifically, we show that visual signals significantly boost audio preference modeling (dropping visual data causes a ~6 pp performance drop in audio), empirically validating the necessity of our training approach.

**Robustness and Text Generalization (Response to R1/gQyH)**: We expand our evaluation to include **HH-RLHF** (text-only) and **ShareGPT-Video** (a diverse video preference benchmark). Results demonstrate that Omni-RRM improves text capability (+2.2 pp over backbone) without degradation and generalizes robustly to non-Qwen video outputs (80.2% accuracy), addressing concerns about backbone limitations and generator-specific bias.

**Comparison with Latest Baselines (Response to R1/gQyH)**: We add direct comparisons with contemporary unified reward models (UnifiedReward-Think and IXC-Reward) across VL-RewardBench, MM-RewardBench, and ShareGPT-Video, where Omni-RRM demonstrates competitive or superior performance while uniquely supporting audio modalities.

Notably, Reviewer gQyH has explicitly acknowledged that our rebuttal successfully addressed their concerns regarding **W3 (Omni Scope), W4 (Baseline Comparisons), W5 (SFT Gap), and W7 (Best-of-N Results)**. We are pleased that Reviewers yzyB, K8o1, and uQjP have maintained their positive scores (6). We believe our comprehensive response, particularly the rigorous validation of our data pipeline and the empirical proof of cross-modal transfer, solidifies the validity of Omni-RRM as a scalable, omni-modal reward modeling approach.

---

### Meta-Review · Area_Chair_4shJ · 2026-01-09

**Summary:**

The paper presents Omni-RRM, a reasoning-driven reward model designed to align Multimodal Large Language Models across text, image, video, and audio. It addresses the scarcity of high-quality preference data through Omni-Preference, a large-scale, automated dataset featuring multi-criteria rationales from teacher models. The model utilizes a two-stage training process— SFT followed by GRPO—to foster interpretable chain-of-thought judgments. Empirical results show state-of-the-art performance, particularly in video and audio domains, significantly outperforming existing open-source baselines and enhancing downstream performance via Best-of-N sampling. The model is open sourced, which is a big plus.

The reviewers widely praised the "zero-human-annotation" approach. By using a "capability-gap" strategy and teacher models to generate rationales, the paper offers a scalable solution to the human-labeling bottleneck. Moving from opaque scalar rewards to structured, chain-of-thought rationales is seen as a significant conceptual advance that improves trust and debuggability. Reviewers also noted the model's impressive SOTA results on video and audio benchmarks, particularly highlighting the value of adding audio-understanding reward capabilities.

Multiple reviewers raised the major limitations lays on the incremental novelty and missing baselines, especially comparing against recent “reasoning as reward" models like UnifiedReward-Think or RM-R1.

Another major question is the "capability-gap" assumption, also raised by multiple reviewers. If a 7B model is paired with a 3B model, the 3B model might often be better/equal, leading to label noise. Reviewer K8o1 specifically noted the lack of statistics on how many samples were discarded during teacher reconciliation.

**Reviewer Concerns:**

The authors actively participated in the rebuttal period and provided more explanations.

*Capability Gap*: The authors elaborated on the assumption that it doesn’t always hold true and introduced a more robust model (e.g., GPT-4o-mini and Doubao-1.5-Pro) to guide the generation of data.

*Novelty*: The authors offered additional clarification of the proposed approach and previous efforts, including UnifiedReward-Think, RM-R1, and others. However, I believe the authors failed to address the central concern here, which is integrating CoT (Common Ground Theory) or thinking into the reward model. Instead, they argued on the SFT -> GRPO pipeline similarity, which I find inaccurate.

**Reviewer Scores:**

Reviewers are likely to maintain the current scores.

---

### Decision · Program_Chairs · 2026-01-26

Reject